# A New Generative Model for Textual Descriptions of Medical Images Using Transformers Enhanced with Convolutional Neural Networks

**DOI:** 10.3390/bioengineering10091098

**Published:** 2023-09-19

**Authors:** Artur Gomes Barreto, Juliana Martins de Oliveira, Francisco Nauber Bernardo Gois, Paulo Cesar Cortez, Victor Hugo Costa de Albuquerque

**Affiliations:** 1Graduate Program in Electrical Engineering, Federal University of Ceará, Fortaleza 60455-760, Brazil; artur.barreto@alu.ufc.br; 2Graduate Program in Teleinformatics Engineering, Federal University of Ceará, Fortaleza 60455-970, Brazil; juliana.martins@unifor.br; 3Controladoria e Ouvidoria Geral do Estado, Governo do Estado do Ceará, Fortaleza 60822-325, Brazil; francisco.gois@cge.ce.gov.br; 4Department of Teleinformatics Engineering, Federal University of Ceará, Fortaleza 60455-970, Brazil; cortez@lesc.ufc.br

**Keywords:** digital image processing, natural language processing, transfer learning, biomedical engineering

## Abstract

The automatic generation of descriptions for medical images has sparked increasing interest in the healthcare field due to its potential to assist professionals in the interpretation and analysis of clinical exams. This study explores the development and evaluation of a generalist generative model for medical images. Gaps were identified in the literature, such as the lack of studies that explore the performance of specific models for medical description generation and the need for objective evaluation of the quality of generated descriptions. Additionally, there is a lack of model generalization to different image modalities and medical conditions. To address these issues, a methodological strategy was adopted, combining natural language processing and features extraction from medical images and feeding them into a generative model based on neural networks. The goal was to achieve model generalization across various image modalities and medical conditions. The results showed promising outcomes in the generation of descriptions, with an accuracy of 0.7628 and a BLEU-1 score of 0.5387. However, the quality of the generated descriptions may still be limited, exhibiting semantic errors or lacking relevant details. These limitations could be attributed to the availability and representativeness of the data, as well as the techniques used.

## 1. Introduction

The field of medicine has undergone a transformative revolution, owing to the rapid advancement of computational technologies. These technological breakthroughs have played a pivotal role in enhancing patient care by enabling more precise and efficient diagnosis, treatment, and patient monitoring. Specifically, the integration of Artificial Intelligence (AI) has emerged as a transformative force, offering promising opportunities to elevate the quality and efficacy of healthcare services. Significant achievements have already been attained through the application of AI in medical contexts. For instance, Ref. [1] developed an automated system for diagnosing COVID-19 and common-acquired pneumonia by utilizing chest CT scans. Additionally, Ref. [2] successfully implemented a multi-class skin lesion detection and classification system through teledermatology.

Generative algorithms have proven to be an efficacious approach for automatically generating textual descriptions of images. These algorithms possess the capability to discern intricate patterns within images and express them coherently in natural language, thereby producing detailed and precise descriptions that faithfully reflect the salient features present in the images.

Within the realm of medicine, the automatic generation of descriptions for medical images has garnered increasing interest due to its potential to assist healthcare professionals in interpreting and analyzing clinical examinations [3]. Elaborate and accurate descriptions of medical images, such as X-rays and magnetic resonance imaging, play an integral role in facilitating accurate diagnoses and effective patient treatment, thereby providing invaluable information to healthcare practitioners. However, the manual generation of such descriptions remains a laborious and error-prone process that necessitates a high level of expertise [4].

Numerous research studies have explored the domain of automatic image description generation in diverse contexts. Nevertheless, there remains a noticeable gap concerning its application to medical images [5]. The distinctive complexities associated with medical images, comprising specific anatomical structures and crucial clinical information, warrant tailored approaches and techniques customized for the medical domain. Hence, there exists a clear need to investigate and develop generative models capable of generating accurate and contextually relevant descriptions for medical images.

Medical images play a crucial role in disease diagnosis, providing physicians with detailed visual information about the internal structures and conditions of the human body. They enable the identification of anomalies, lesions, or pathological changes, even at early stages, before visible or palpable symptoms appear [6]. For example, X-rays, computed tomography (CT) scans, magnetic resonance imaging (MRI), and ultrasounds can detect early changes in diseases, allowing for early detection and increasing the chances of effective treatment and better outcomes for patients. Moreover, medical images help identify and characterize lesions, such as tumors, fractures, inflammations, and congenital abnormalities. They provide essential information about the size, location, extent, and characteristics of these lesions, enabling physicians to determine the most suitable treatment approach.

In this context, the development of generative models capable of automatically and efficiently producing descriptive texts for medical images becomes crucial. Previous research has explored various approaches for automatic description generation, including rule-based models, statistical models, and neural network-based models [7]. These approaches have shown promising results in different domains, encompassing both general image description and specific medical image description. However, despite recent advancements, automatic description generation for medical images still faces significant challenges due to the complexity and diversity of medical images.

The potential of automatic description generation for medical images lies in optimizing the workflow of healthcare professionals, facilitating more efficient and accurate analysis of clinical exams [8]. Implementing such solutions could lead to substantial benefits for patients and healthcare providers, streamlining the decision-making process and enhancing the overall quality of medical care [9].

The objective of the present work is to address the challenge of implementing a generative model for medical image descriptions, with a specific focus on its application to the publicly available "Radiology Objects in Context" (ROCO) dataset [10]. The central research question seeks to explore the effective integration of natural language processing techniques and machine learning to generate precise and contextually relevant descriptions for medical images within the ROCO dataset.

The methodological approach adopted in this study consists of a combination of natural language processing and machine learning techniques. A strategy is used where image recognition models are employed to extract relevant attributes from medical images. These attributes are used as inputs to a generative model based on neural networks that create contextually relevant textual descriptions for their inputs. The model training is performed using an annotated dataset, where each image is associated with a corresponding textual description. The model evaluation is performed through various metrics and a comparison with the provided real descriptions.

The main contrubution of this work are:Evaluate and compare the performance of different specific models concerning the quality of the generated descriptions;Explore the generalization of models to different medical image modalities, such as X-rays, MRI, or CT scans, and to a variety of medical conditions.

## 2. Materials and Methods

In this section, all computational techniques used to generate textual descriptions of medical images are presented. Firstly, the ROCO dataset is introduced. Then, the algorithm for selecting a subset of images and texts from this database is described. Next, the experiment to investigate the possible combinations of the studied variables is explained, aiming to obtain the best combination of analyzed techniques. Subsequently, the generative algorithm built using deep learning and natural language techniques for generating descriptions of medical images is presented. Finally, the parameters used in the performance evaluation of the generative algorithm are described.

### 2.1. The ROCO Dataset

In this study, the Radiology Objects in Context (ROCO) dataset was utilized. This dataset comprises approximately 82,000 radiology images from diverse modalities, including computed tomography, ultrasound, X-rays, fluoroscopy, positron emission tomography, mammography, magnetic resonance imaging, angiography, among others. Figure 1 shows an example subset of these images.

In addition to radiological images, the dataset also contains medical images of other types, such as figures, portraits, digital arts, illustrations, and clinical photos. This set could potentially enhance prediction and classification performance; however, it was not utilized in this study. Figure 2 displays a random subset of these images.

Each image in the ROCO dataset is associated with various text files containing corresponding image descriptions, keywords characterizing it, two UMLS (Unified Medical Language System) codes, a CUI (Concept Unique Identifier), SemTypes (Semantic Types), and their respective download links. Figure 3 provides random images from the dataset and their respective three-word descriptions.

Using the ROCO dataset, it is possible to create systems for image description generation, allowing for multimodal representation for image sets that do not have a textual representation, which is the main objective of this work. Additionally, it is possible to create systems with the aim of structuring images and providing semantic annotation for image retrieval and information purposes.

### 2.2. Training, Testing, and Validation Data Sets

As presented in the previous section, the ROCO database contains approximately 82,000 medical images from different contexts. Therefore, to achieve the goal of creating natural language sentences that describe the content of medical images through generative models, the first step was to create an algorithm capable of selecting a subset of image-text pairs from a well-defined context within the ROCO database.

Firstly, an algorithm that calculates the frequency of all keywords of the dataset in descending order was developed. A portion of this output can be seen in Figure 4.

As there are 37,075 distinct keywords with frequencies ranging from 1 to 14,870, an arbitrary lower limit for word frequency was set, selecting only those with a frequency greater than 500. This way, only the most relevant words from the ROCO dataset are included.

Analyzing the list of keywords with a frequency greater than 500, it was possible to group most of them into three distinct categories. Words that could not be classified into any of these categories were discarded. The categories are:Type of examination;Body part;Identified problem.

These words belonging to the three defined categories were designated as keywords of interest. Table 1 shows the keywords of interest for each of the defined categories and their frequencies in parentheses.

Then, the algorithm generates a list of all possible combinations, taking three of the values of interest defined in each category at a time. For each combination of keywords of interest, the algorithm searches for images that match these values and saves them in a specified directory.

This algorithm iterated over the list of all possible combinations and separated groups of ROCO images belonging to combinations of keywords that contained between 10 and 80 images (arbitrary defined size range of a subclass). This resulted in a dataset containing 1419 medical images and their respective descriptive texts. The texts and the quantity of images for each group can be found in Table 2.

Then, this set of 1419 medical images was randomly divided into three smaller parts, in the proportion of 60%/20%/20%, consisting of the training, validation, and test sets for the generative models implemented in this study.

### 2.3. Description of the Experiment

In this subsection, the experiment conducted to create a generative model of descriptive texts for medical images is described in detail. The main objective of the experiment was to investigate the possible combinations of the studied variables to gain valuable insights into the model’s performance.

Firstly, it is important to emphasize that all the experiments were conducted on a Dell Alienware M15 R7 notebook with the following configurations: 12th Generation Intel Core i7-12700H, 2.30 GHz, 20 cores; 32 GB; 1 TB SSD; NVIDIA GeForce RTX 3070 Ti with 8 GB dedicated memory; Windows 11.

To extract attributes from the images, a variety of classical image recognition models were used: MobileNetV2 [11], DenseNet201 [12], ResNet152V2 [13], NASNetLarge [14], VGG19 [15], Xception [16], InceptionV3 [17], and InceptionResNetV2 [18]. Each model was individually employed to evaluate its ability to capture relevant information from the medical images.

Furthermore, different gradient optimizers were explored for model training: Adam [19], AdamW [20], Adadelta [21], and Adafactor [22]. This aimed to determine which optimizer would provide the best results in terms of computational performance and model convergence.

Another line of experimentation involved the use of transfer learning in the image feature extractors. Therefore, in part of the experiments, feature extractors pre-trained with the ImageNet dataset were used. Additionally, regarding transfer learning, cases with and without weight fine-tuning during the training phase were tested to evaluate the influence of this process on the results. Experiments without using ImageNet transfer learning were also conducted to explore the model’s ability to learn directly from the available medical data.

During the experiment, data augmentation techniques, known as image augmentation, were also considered. These techniques aim to expand the available dataset by applying transformations such as rotation, zoom, and mirroring to the images. Experiments were conducted with and without the use of data augmentation to analyze its impact on the model’s performance.

It is important to note that all hyperparameters for each test section were optimized using the KerasTuner API with the Grid Search approach. This process allowed for a systematic exploration of hyperparameter combinations and aided in identifying the best configurations for each model.

During the training of each variant, model performance was monitored through accuracy, as well as the processing time involved, measured by the time required for training and testing. The best model is defined as the one with the highest accuracy.

To address possible limitations of the proposed solution, imbalanced data was added to the training, validation, and test sets. This choice allowed for an analysis of the impact of class imbalance on the model’s performance and investigated the minimum dataset size required to achieve satisfactory results. This analysis was essential to provide insights into the model’s ability to effectively handle imbalanced data.

Finally, for the best obtained model, a detailed evaluation of its performance was conducted by constructing confusion matrices for each part of the generated texts (exam type, body part, and identified problem). These matrices allowed for visualizing the correct and incorrect choices made by the best model in constructing the descriptive texts of medical images. Key indicators, such as precision, recall, F1 score, and others, were calculated for each matrix. These indicators provided an objective measure of the quality of the textual descriptions generated by the model and served to evaluate its performance.

By combining all the studied variables and conducting systematic experiments, various combinations of feature extraction models, gradient optimizers, use of transfer learning (with and without weight fine-tuning), data augmentation techniques, and the presence of imbalanced data were explored. With this approach, it was possible to gain an in-depth understanding of the performance, processing time, and limitations of the proposed solution.

### 2.4. Proposed Generative Model

For text generation, transformers stand out, which are machine learning models based on attention mechanisms. Instead of sequentially processing words, e.g., how RNNs operate, transformers calculate interactions between all words in a sentence using attention operations [23]. These models have been successfully applied in generating descriptions for medical images because they have the ability to capture complex relationships between words and generate more contextually relevant descriptions [24].

Our generative text model consists of two interconnected transformer blocks: a transformer encoder block and a transformer decoder block.

In Figure 5, a summarized representation of the proposed generative model for medical descriptions is depicted. In subsequent paragraphes, each element in this representation will be explored. The code for the proposed model can be accessed at https://github.com/ArturBarreto/GenerativeModelTextualDescriptionsMedicalImages (accessed on 6 August 2023).

The transformer encoder block is used to encode the features extracted by the CNN from the medical images and generate a vector representation of this information. To build this encoder, a class named “Transformer Encoder Block” was defined. This encoder comprises attention layers, normalization layers, and a dense layer, which collectively transform the input into a more meaningful representation for the model.

First, the input is normalized and then passed through the “Dense” layer. This layer consists of a fully connected dense layer with a ReLU activation function. The result is then passed through the “Multi-Head Attention” layer with multiple heads, which returns an “attention utput” tensor. This tensor is added to the original input and normalized by the “Layer Normalization” layer. The final result is a vector representation of the image attributes fed into the generative model. This vector is returned as the output of the transformer encoder.

The transformer decoder block operates by taking as input a tensor containing text tokens and the outputs generated by the transformer encoder block. It consists of multiple layers, including Multi-Head Attention layers, normalization layers, and dense layers. The decoder processes the input tokens, ensuring that each token receives the relevant contextual information from other tokens through Multi-Head Attention. It also incorporates positional encoding to account for the sequential order of tokens.

This block aims to generate a probability distribution over the vocabulary, effectively producing the next token in the sequence. The hyperparameters, such as the dimension of word embeddings, the number of attention heads, and vocabulary size, play crucial roles in determining the quality and coherence of the generated text.

This combination of components forms a strong foundation for a generative system that can convert medical image data into understandable text, improving interpretability and usefulness in medical image analysis.

### 2.5. Performance Evaluation Metrics

In this subsection, the metrics used to evaluate the performance of the proposed model are presented.

Accuracy measures the proportion of correct words generated compared to the total number of words generated and can be calculated using the equation:(1)$$Acc=\frac{\mathrm{Number}\;\mathrm{of}\;\mathrm{correct}\;\mathrm{words}\;\mathrm{generated}}{\mathrm{Total}\;\mathrm{number}\;\mathrm{of}\;\mathrm{words}\;\mathrm{generated}}.$$

This equation is commonly used in the context of natural language processing (NLP), machine learning, and text generation tasks to evaluate how well a system is performing in terms of generating correct words and is expressed as a percentage or a decimal between 0 and 1. It measures the proportion of correctly generated words out of the total words generated by the system.

The loss represents the measure of the error between the model’s word predictions and the actual words in the dataset. In this work, the loss is calculated using Sparse Categorical Cross-Entropy. This function is commonly used in multi-class classification tasks, where true labels are represented by integer values, i.e., in sparse encoding, and the model’s predicted probabilities are calculated using a softmax activation function to obtain a probability distribution over the classes [25]. The Sparse Categorical Cross-Entropy aims to minimize the difference between the model’s predicted probabilities and the true labels, penalizing incorrect classifications more heavily, and can be calculated using the equation:(2)$$\mathrm{Loss}\left(y,\hat{y}\right)=-\sum_{i=1}^{C}y_{i}\ln\left({\hat{y}}_{i}\right).$$

The Sparse Categorical Cross-Entropy loss function is a well-suited choice for training generative text models for several compelling reasons. First and foremost, it is designed to handle multi-class classification problems, making it appropriate for generating text where each word or token corresponds to a specific class in the vocabulary. This loss function efficiently quantifies the dissimilarity between the predicted probability distribution over classes and the true class labels, promoting the generation of more accurate and contextually relevant text. Additionally, it naturally accommodates the sparsity of the target distribution in text generation tasks, as most words in a given language are not present in any given context. This makes it especially valuable for modeling realistic language generation scenarios where words are drawn from a vast vocabulary.

The accuracy metric provides an overall measure of the model’s ability to generate correct words, while the loss metric indicates how close the model’s predictions are to the ground truth during training. A lower value of the loss indicates better alignment between predicted and true labels, reflecting better performance.

F1 score is a measure that combines precision and sensitivity into a single metric. It is the harmonic mean between these two metrics and provides a balance between them. It can be calculated using the equation:(3)$$F1\;\mathrm{score}=2\times \frac{\mathrm{Precision}\times \mathrm{Sensitivity}}{\mathrm{Precision}+\mathrm{Sensitivity}}.$$

Additionally, for the best model, the BLEU-1 metric was calculated. BLEU-1 is a commonly used metric to evaluate the quality of translations or text generation. It measures the overlap of unigrams between the generated sequences and the reference sequences. It can be calculated using the equation:(4)$$\mathrm{BLEU}-1=\frac{\sum_{1-\mathrm{grams}}\mathrm{number}\;\mathrm{of}\;\mathrm{correctly}\;\mathrm{predicted}\;1-\mathrm{grams}}{\sum_{1-\mathrm{grams}}\mathrm{total}\;\mathrm{number}\;\mathrm{of}\;\mathrm{predicted}\;1-\mathrm{grams}}.$$

## 3. Results

This section presents and discusses the results obtained from a series of experiments conducted to create a generative model of descriptive texts for medical images. The outcomes are organized in tables, each representing a specific experiment configuration involving different combinations of image feature extraction models, gradient optimizers, transfer learning usage, and data augmentation.

To facilitate the interpretation of the results, the columns in the tables are defined as follows:**ID**: a unique identifier for each experiment configuration;**OPTZ**: the gradient optimizer used in the model training;**TL**: an abbreviation for Transfer Learning, indicating whether the concept of transfer learning from ImageNet was utilized;**TR**: an abbreviation for Trainable, indicating whether the model weights were fine-tuned during the training stage;**IA**: an abbreviation for Image Augmentation, indicating whether a data augmentation technique was applied to the images;**Loss**: the final value of the loss function obtained by the model;**ACC**: the final accuracy achieved by the model;**Epochs**: the number of training epochs;**Training Time (s)**: the time required for model training in seconds;**Test Time (s)**: the time required for model testing in seconds.

These metrics provide valuable information regarding the models’ performance in terms of loss, accuracy, and training/testing time. By analyzing this information, it becomes possible to compare different experiment configurations and identify those that achieved the best results.

The provided data allows for an evaluation of the models’ performance concerning various combinations of tested variables, such as feature extractors, gradient optimizers, the use of transfer learning, and data augmentation techniques.

It is important to note that, after conducting all 192 experiments, the mean (μ) and standard deviation (σ) of accuracy were calculated, resulting in the values presented in Table 3. These results indicate that the experiments show statistically significant differences regarding accuracy. Therefore, considering only this metric, it is possible to define the best model as the one with the highest accuracy.

### 3.1. DenseNet201 Family

Upon evaluating the results in Table 4, considering the various variables presented, several significant trends can be identified.

Firstly, different gradient optimizers exhibit considerable impact on the model’s performance. Notably, the Adafactor optimizer consistently demonstrated lower losses and higher accuracies across various settings. This suggests that Adafactor is an effective optimizer for training the DenseNet201 model in this specific context.

Furthermore, the use of transfer learning generally improved results when compared to settings without transfer learning. Configurations with TL achieved lower losses and higher accuracies across various gradient optimizer combinations. This emphasizes the importance of leveraging pre-trained knowledge from broader models and adapting them to specific tasks.

It is worth noting that the best results in this family occurred in cases where the model’s weights were fine-tuned during the training stage. This implies that while ImageNet can serve as a good starting point for these weights, better results can be achieved by fine-tuning the weights during training.

Concerning data augmentation, configurations with image augmentation consistently showed lower losses and higher accuracies, indicating that data augmentation was beneficial for the model’s performance. This technique improves model generalization, enhancing robustness and reducing overfitting.

Additionally, the configurations exhibited significant variations in training times, ranging from a few hundred seconds to several hours. These differences can be influenced by various factors, such as dataset size, model complexity, among others. In general, configurations with transfer learning and image augmentation tend to require more training time due to additional processing needed for weight adjustments or data transformations. Conversely, configurations without transfer learning and without image augmentation may have shorter training times as they deal solely with the original dataset and do not involve complex model adjustments.

When evaluating the results in the tables, several factors should be considered to determine the best outcome. Accuracy (ACC) serves as an important indicator of the model’s performance, measuring the proportion of correct predictions out of the total predictions. Additionally, other metrics such as loss, F1 score, precision, and recall can be taken into account.

Upon analysis, some configurations achieved relatively high accuracy. For instance, configuration ID 20 achieved an accuracy of 0.7610, which is the highest among the tested configurations. This configuration employed the DenseNet201 architecture as a feature extractor for medical images, the Adafactor optimizer, transfer learning with weight adjustment, and image augmentation.

Furthermore, other criteria, such as processing time and model scalability, need to be considered. Some configurations that attained high accuracy may require significantly longer training time, which could be impractical in certain cases. Thus, finding a balance between performance and processing time is important, taking into account the available resources and project constraints.

Based on the presented results, configuration ID 20 appears to be one of the most promising ones, as it achieves high accuracy and involves techniques such as transfer learning, image augmentation, and weight adjustment. However, it is essential to conduct further analyses and consider the specific requirements of the application or task at hand before finalizing the choice.

### 3.2. ResNet152V2 Family

When analyzing the results from Table 5 and considering the various variables present in the table, several relevant trends can be identified.

Table 5 presents the results of experiments conducted with the ResNet152V2 model using different configurations. Each row represents a specific configuration identified by its ID number, and the columns show various metrics such as optimizer (OPTZ), transfer learning (TL), weight adjustment during training (TR), data augmentation (IA), loss, accuracy (ACC), number of epochs, training time, and test time.

Regarding the optimizers, we observed that the AdamW optimizer (IDs 26, 30, 34, 38) tends to yield good results in terms of accuracy, with values above 0.72, and relatively low losses. Similarly, the Adafactor optimizer (IDs 28, 32, 36, 40) also shows promising results in terms of accuracy, with values above 0.73.

The use of transfer learning (TL) and data augmentation (IA) techniques can lead to significant improvements in the model’s performance. Several configurations that apply these techniques show higher accuracy compared to configurations without them. For example, configurations with IDs 29, 33, 37, and 41 exhibit superior results when both transfer learning and data augmentation are applied.

It is worth noting that some configurations, including transfer learning and/or data augmentation, can lead to an increase in training time. This is because these techniques often involve using pre-trained models or generating more training examples, which require more processing time. Therefore, when considering these techniques, it is necessary to strike a balance between model performance and the required training time. However, the test time does not seem to be significantly affected by the evaluated configurations.

The results indicate that the use of specific optimizers such as Adafactor and AdamW, combined with transfer learning, can lead to promising performance in terms of accuracy. The best result in Table 5, considering the highest achieved accuracy, belongs to configuration ID 36. This configuration achieved an accuracy of 0.7359 after nine training epochs. In this configuration, the ResNet152V2 model was used with the Adafactor optimizer, and transfer learning (TL) was applied.

Additionally, it is relevant to consider the training and testing time associated with this configuration. In Table 5, it is indicated that the training time was 3771 s (approximately 1 h and 3 min), and the testing time was 452 s (just under 8 min). Therefore, it is essential to consider the achieved accuracy along with the training and testing time to evaluate the efficiency and practicality of this configuration in a real-world context.

### 3.3. NASNetLarge Family

Table 6 presents the results of experiments conducted with the NASNetLarge model using different configurations.

Four optimizers were tested during the training of the models: Adam, AdamW, Adadelta, and Adafactor. Each optimizer has its own weight adjustment characteristics and learning rates. The results show that there is no significant difference in the accuracy achieved by the different optimizers, ranging from approximately 0.69 to 0.72.

Regarding the use of pre-trained transfer learning with the ImageNet dataset, the results indicate that its use tends to provide slightly better accuracies compared to experiments without transfer learning. This suggests that transferring knowledge from pre-trained models can assist in learning relevant features for medical images.

In addition to transfer learning, weight adjustment during training was evaluated. The results show that weight adjustment can also contribute to slightly better performance, with slightly higher accuracies compared to experiments without weight adjustment. This technique allows the model to adapt better to the specific data of the training set.

Regarding the use of data augmentation techniques, the results indicate that its use does not show a clear trend of improvement or deterioration in accuracy. This suggests that, for the NASNetLarge model, data augmentation may not be as crucial for performance.

It is important to note that experiments with more complex configurations, such as the use of transfer learning, weight adjustment, and data augmentation techniques, generally require longer training time. This should be considered when implementing the model in a production environment, where training and testing time are important factors.

Overall, the results of the experiments with the NASNetLarge model indicate that the use of transfer learning, weight adjustment, and data augmentation techniques can contribute to slightly better performance, resulting in slightly higher accuracies. However, the choice of these techniques should take into account the processing time involved, as more complex configurations may require longer training time.

Among the experiments conducted in the NASNetLarge family, the best result was achieved in the configuration identified as ID 68. These results indicate that this configuration performed well in the generative model of descriptive texts for medical images. The loss, which measures the discrepancy between the model’s predictions and the true labels, showed a relatively low value, indicating a good model fitting capacity to the data. The configuration achieved these results after seven training epochs, with a total training time of 4260 s and a testing time of 768 s.

### 3.4. VGG19 Family

Table 7 presents the results of different configurations of the model using the VGG19 convolutional neural network (CNN).

In relation to the optimizers, it was observed that configurations with Adam and AdamW generally show more favorable loss and accuracy values compared to configurations with Adadelta and Adafactor.

It was found that, in general, the use of transfer learning and weight adjustment during the training stage improved the model’s performance, as evidenced by the configurations marked with ‘√’ in those columns. The use of data augmentation techniques (IA) also shows an impact on the results. Configurations with data augmentation tend to have lower loss values and higher accuracy compared to configurations without data augmentation.

Regarding the training and testing time, there are significant variations among the different configurations. The training time can range from just over 1800 s to over 9000 s, while the testing time varies from about 150 s to over 300 s. In summary, the analysis of Table 7 indicates that using VGG19 together with the Adam optimizer and data augmentation techniques can lead to good performance in the task of generating descriptive texts for medical images. Transfer learning and weight adjustment also contribute to improving the model’s performance.

The best result obtained in Table 7 is represented by configuration with ID 85. In this configuration, the VGG19 architecture was used together with the Adam optimizer, as well as the combination of transfer learning and data augmentation.

This specific configuration achieved a loss of 0.6276 and an accuracy of 0.7628. These values indicate very favorable performance in the task of generating descriptive texts in medical images.

The application of transfer learning made it possible to leverage prior knowledge learned by the VGG19 network on a related task. This can lead to better performance since the network has already been trained on a large amount of general image data.

Furthermore, the use of data augmentation techniques contributes to increasing the quantity and diversity of the training data. This can improve the model’s ability to generalize and handle variations in medical images, resulting in more robust performance.

The Adam optimizer is known for its efficiency and effectiveness in training neural networks, adapting the learning rate adaptively for each model parameter. This feature may have contributed to the superior performance achieved in this configuration.

In conclusion, the best result obtained in Table 7, represented by configuration with ID 85, demonstrates the importance of combining different strategies, such as transfer learning, data augmentation, and efficient optimizers, to achieve remarkable performance in generating descriptive texts in medical images.

### 3.5. Xception Family

The Table 8 presents the results of experiments conducted with the Xception model, employing different gradient optimizers, transfer learning, data augmentation, and weight adjustment during training.

When analyzing the results, it becomes evident that the choice of gradient optimizer has a notable impact on the model’s performance. For instance, the configuration using the Adam optimizer achieved a loss of 0.7686 and an accuracy of 0.7215, while the configuration employing the Adadelta optimizer obtained a loss of 0.7416 and an accuracy of 0.7334. This variation suggests that selecting an appropriate optimizer is crucial for achieving optimal model performance.

Configurations with transfer learning generally showed slightly improved results in terms of loss and accuracy compared to configurations without transfer learning. For example, the configuration with transfer learning using the Adam optimizer had a loss of 0.7961 and an accuracy of 0.7069, while the corresponding configuration without transfer learning achieved a loss of 0.7521 and an accuracy of 0.7235. This indicates that leveraging pre-trained resources, such as feature extractors from the ImageNet dataset, enhances the model’s ability to capture relevant information from medical images.

Another critical factor is the application of data augmentation techniques. Comparing configurations with and without data augmentation, it can be observed that, in some cases, data augmentation leads to improved model performance. For instance, the configuration with the Adafactor optimizer and data augmentation (IA) yielded a loss of 0.7381 and an accuracy of 0.7289, whereas the corresponding configuration without data augmentation showed a loss of 0.7687 and an accuracy of 0.7131. This suggests that data augmentation aids the model in learning more robust patterns and generalizing better.

The training and testing times recorded in Table 8 provide insight into the processing time associated with each configuration. It is evident that the time varies significantly among configurations, highlighting the importance of considering the trade-off between model performance and computational resources required.

Overall, the analysis of these results emphasizes the importance of exploring different combinations of variables, such as feature extractors, gradient optimizers, the use of transfer learning, and data augmentation techniques, to gain a deeper understanding of the performance and limitations of the generative model for descriptive texts in medical images.

The best model in this family, represented by ID 111, employed the Xception architecture, the Adadelta optimizer, transfer learning, and data augmentation techniques. This model demonstrated good overall performance with a relatively low loss of 0.6895 and a considerably high accuracy of 0.7499, indicating its ability to learn relevant patterns in medical images and make accurate predictions. The training time for this model was 1238 s, emphasizing the importance of considering available computational resources.

Based on the presented results, this model shows potential for generating accurate and high-quality textual descriptions for medical images.

### 3.6. InceptionV3 Family

Table 9 presents the results of different configurations used in the experiment for the InceptionV3 model. Observing the results, several key observations and discussions can be made.

The AdamW, Adadelta, and Adafactor optimizers demonstrated similar accuracies, with slightly better results than the Adam optimizer. In terms of training and testing time, the AdamW and Adafactor optimizers exhibited shorter times compared to Adam and Adadelta.

The use of transfer learning and data augmentation had a positive effect on the overall accuracy of the models. The results indicate that combining transfer learning and data augmentation led to higher accuracies compared to when these techniques were used separately. However, it is important to note that data augmentation may lead to an increase in training time. This suggests that using pre-trained models and data augmentation techniques is beneficial for improving the performance of the descriptive text generation model for medical images.

The best model in this family, identified by ID 136, achieved a loss of 0.7118 and an accuracy of 0.7472, indicating its capability to generate accurate textual descriptions for the medical images used in the experiment. The configuration of the best model included the use of transfer learning with the InceptionV3 architecture, which is a pre-trained model with the ImageNet dataset. This approach effectively leverages prior knowledge acquired by the model from a large set of images and applies it to a new domain, such as medical images. This allows the model to capture relevant information from medical images and use it to generate more accurate textual descriptions. Additionally, the model also benefited from data augmentation techniques, which helped improve its ability to generalize to different variations in medical images.

In terms of computational performance, the model was trained for eight epochs, and the training time was 1059 s (approximately 17 min). The testing time was 187 s (about 3 min). These times are essential to assess the efficiency of the model in terms of processing time.

Considering all these factors, it can be concluded that the model with ID 136 showed good performance in generating descriptive texts for medical images. The combination of transfer learning with the InceptionV3 architecture, the Adafactor optimizer, and data augmentation techniques allowed the model to capture relevant information from the images, resulting in accurate textual descriptions with an accuracy of 74.72%.

### 3.7. InceptionResNetV2 Family

After careful analysis of the results presented in Table 10, it is evident that the performance of the InceptionResNetV2 model varies depending on the combinations of gradient optimizers, the usage of transfer learning, data augmentation techniques, and weight adjustment during training. However, no clear performance pattern was identified with respect to these variables.

The impact of different gradient optimizers on the model’s accuracy and loss was not significantly different. The Adam and AdamW optimizers showed consistent performance across various configurations, with similar results in terms of accuracy and loss. On the other hand, the Adadelta and Adafactor optimizers achieved slightly lower accuracy in certain configurations.

Regarding the use of transfer learning, it did not have a significant impact on the model’s performance. Configurations with and without transfer learning achieved similar results in terms of accuracy and loss.

The inclusion of weight adjustment during training did not show a clear advantage or disadvantage. Configurations with and without weight adjustment exhibited comparable performance in terms of accuracy and loss.

The application of data augmentation techniques resulted in a small increase in accuracy in some configurations, but this improvement was not consistently observed. In certain setups, data augmentation led to enhanced accuracy, while in others, there was no significant improvement. Overall, the impact of data augmentation on the model’s performance was moderate.

It is important to note that the training and testing times varied considerably for each configuration. As this family of models demonstrated very similar results, the computational efficiency may be a relevant factor when choosing the appropriate gradient optimizer and other techniques.

In conclusion, the InceptionResNetV2 family obtained consistent results in terms of accuracy and loss compared to other tested models, and no clear superior or inferior model was identified concerning the analyzed variables.

### 3.8. MobileNetV2 Family

The results of different configurations used in the experiment for the MobileNetV2 model are presented in Table 11. Upon analyzing these results, the following observations and discussions can be made.

The AdamW gradient optimizer consistently showed better results in terms of loss and accuracy, especially when used in combination with transfer learning and weight adjustment during training.

The utilization of transfer learning, with pre-trained feature extractors from the ImageNet dataset, tended to slightly improve the model’s accuracy, particularly when combined with weight adjustment during training. This indicates that the prior knowledge acquired from ImageNet can be effectively transferred to the task of generating descriptive texts for medical images, and adjusting the weights of the feature extractors during training can aid the model in learning specific characteristics of medical images.

Regarding data augmentation, the results indicated that in terms of loss, experiments with data augmentation generally had slightly higher values compared to experiments without data augmentation. This suggests that data augmentation might introduce some variability in the results and increase the model’s complexity. However, the difference in loss was not significant.

In terms of accuracy, data augmentation showed mixed results. In certain configurations, data augmentation led to an increase in accuracy compared to experiments without it, while in others, there was no significant improvement.

It is crucial to consider that while data augmentation might increase accuracy in some cases, it could also lead to longer training times. The experiments with data augmentation generally required more training time compared to those without it.

In summary, data augmentation can have a positive impact on the model’s accuracy in specific configurations, but the trade-off between increased accuracy and longer training time needs to be considered.

The best model of the MobileNetV2 family, identified by ID 191 in Table 11, corresponds to the configuration with the Adadelta optimizer, the use of transfer learning, and data augmentation. This best model achieved a loss of 0.7261 and an accuracy of 0.7345, demonstrating good performance in generating descriptive texts for medical images.

Additionally, the model’s fast convergence during training, completed in just four epochs, can be attributed to the effective combination of the Adadelta optimizer with other model variables. In terms of training and testing time, the model required about 400 s (approximately 6 min and 40 s) for training and 93 s (about 1 min and 33 s) for testing. These times are considered reasonable and indicate adequate computational efficiency.

In conclusion, the best model (ID 191) demonstrated promising results in terms of loss, accuracy, and training time, making it a potential configuration for the task of generating descriptive texts for medical images.

### 3.9. Best Results from Each Family

Table 12 provides a summary of the experiment’s best results from each family of models used to extract attributes from medical images. It is important to note that the InceptionResNetV2 family did not have a clearly defined best result. This table enables a direct comparison between different model families and their respective performances, allowing us to observe the relevance of various approaches in achieving the best result.

The Adafactor optimizer was utilized in four out of the seven experiments with the best results (IDs 20, 36, 68, and 136), while the Adadelta optimizer was used in two experiments (IDs 111 and 191). Only one experiment employed the Adam optimizer (ID 85). These results suggest that using the Adafactor optimizer may be associated with better performances, as it appears in four of the best results. However, it is crucial to acknowledge that other factors, such as the architecture of the attribute extractor, the use of transfer learning, weight adjustment during training, and the employment of data augmentation, can also influence the results.

All the best results relied on transfer learning with a pre-trained network on the ImageNet dataset, highlighting the importance of this technique for the overall performance of the models. This indicates that initializing the networks with pre-trained weights contributed to superior results due to the prior learning on image attribute extraction tasks.

Regarding weight adjustment during training, out of the seven best results listed, three of them (IDs 20, 68, and 191) indicated the use of this technique. This information suggests that weight adjustment may not have been a significantly relevant strategy to improve the models’ performance, as the weights initialized via transfer learning with pre-trained weights from the ImageNet dataset already demonstrated sufficient capability to extract attributes from medical images.

As for data augmentation, it was present in four out of the seven experiments (IDs 85, 111, 136, and 191). This indicates that data augmentation likely played an important role in enhancing the models’ performance, as it can improve the model’s generalization and ability to handle variations in medical images, resulting in more robust performance.

Finally, the training and testing times varied significantly among the different models. These times can be influenced by various factors, such as the network’s complexity, the type of gradient optimizer used, and the application or absence of data augmentation techniques. Training times ranged from 400 s (ID 191-MobileNetV2) to 8930 s (ID 85-VGG19), while testing times ranged from 93 s (ID 191-MobileNetV2) to 768 s (ID 68- NASNetLarge). It is important to note that training and testing time do not always directly correlate with the best performance achieved by the models.

The best result obtained across all families, considering the highest accuracy, corresponds to variant ID 85. This variant consists of a VGG19 model trained using transfer learning and data augmentation. The achieved loss value was 0.6276, and the obtained accuracy was 0.7628.

Based on the data available in Table 12, it is evident that the VGG19 model achieved a very high level of accuracy, indicating its ability to generate descriptive texts for medical images effectively.

The utilization of transfer learning, combined with data augmentation, likely contributed to the VGG19 model’s exceptional performance. Transfer learning enabled the model to benefit from prior knowledge learned in image attribute extraction tasks, while data augmentation increased the variability of the training data, leading to a more robust and generalizable model.

However, it is crucial to emphasize that for a comprehensive analysis of the best model’s performance, additional metrics such as precision, recall, and F1 score are necessary to conduct statistical analyses and determine the significance of the results. Such analyses will be the focus of the subsequent section.

## 4. Discussion

The best model obtained in the experiment underwent a detailed evaluation of its performance through the construction of confusion matrices for each part of the generated texts: type of examination, body part, and identified problem.

Figure 6 provides an example representing two arbitrary images from the test dataset along with their respective true descriptive texts and the texts generated by the best model, which achieved an accuracy of 0.7628 and a BLEU-1 score of 0.538.

The utilization of confusion matrices allowed a visualization of both the correct and incorrect choices made by the best model in constructing each part of the descriptive texts for the medical images. From these matrices, essential indicators were calculated to provide objective measures of the textual descriptions’ quality generated by the model, facilitating the evaluation of its performance. As such, each confusion matrix has an associated table summarizing these indicators.

The confusion matrix is a valuable tool for evaluating the performance of a classification model, displaying the model’s predictions concerning the actual classes. In the confusion matrices presented in this work, the actual classes are represented on the vertical axis, while the predicted classes are represented on the horizontal axis.

### 4.1. Confusion Matrix for the Type of Examination

The confusion matrix represented in Table 13 illustrates the best model’s predictions regarding the type of exam for the images. Observing this confusion matrix allows for several conclusions to be drawn.

The model’s performance for the ‘ct’ (computed tomography) class is quite satisfactory, with 108 examples correctly classified. There were only a few confusions, with two examples misclassified as ‘radiograph,’ ten as ‘scan,’ and three as ‘tomography.’

The ‘radiograph’ class also demonstrated good performance, with 32 examples correctly classified. However, there were some confusions, particularly with the ’xray’ class, where 12 radiograph examples were misclassified. Nevertheless, it is worth noting that ’xray’ and ‘radiograph’ are considered interchangeable terms in this context.

The model’s performance for the ‘scan’ class was not as strong, with 26 examples correctly classified. There were confusions, with 19 examples being misclassified as ‘ct’ and five examples as ‘tomography’.

The ‘tomography’ class exhibited poor performance, with no examples correctly classified. There was confusion, with nine examples being mistaken for the ‘scan’ class and 16 examples being misclassified as ‘CT.’ Nevertheless, ‘tomography’ and ‘computed tomography’ can be considered interchangeable terms in this context.

On the other hand, the ‘xray’ class demonstrated excellent performance, with 11 examples correctly classified. There were no confusions with other classes in this category.

For the ‘angiogram’ and ‘angiography’ classes, both showed satisfactory results, with most examples correctly classified. However, seven examples were misclassified as ‘CT.’

The ‘abdominal’ class had poor performance, with no examples being correctly classified. There were confusions, with four examples being mistaken for the ‘ct’ class and one example being misclassified as ‘scan.’

The ‘contrast’ class also had poor performance. All examples were misclassified as ’angiogram,’ with no confusions with other classes.

Table 14 presents the metrics of the confusion matrix for the best model regarding the type of examination in the images. Analyzing this table allows an evaluation of the best model’s performance and provides insights into its effectiveness.

Upon analyzing the metrics of the confusion matrix for all provided classes, considerable variation in the results is evident. Let us discuss each class in a general context and assess whether the values are considered good or poor.

For the ‘ct’ class, the performance is relatively good. Sensitivity and specificity are above 0.7, indicating that the model can correctly identify most positive and negative cases. The precision is also around 0.71, indicating that the majority of cases classified as positive are indeed positive.

Regarding the ‘radiograph’ class, the performance is quite positive. Sensitivity and specificity are above 0.7 and 0.9, respectively, indicating good performance in correctly identifying positive and negative cases. The precision is high, around 0.94, suggesting that most positive classifications are correct. Additionally, the F1 score of 0.81 indicates a good balance between precision and sensitivity.

For the ‘scan’ class, the values are slightly lower compared to the previous classes. Sensitivity and specificity are above 0.5, indicating reasonable performance in identifying positive and negative cases. However, precision and the F1 score are around 0.57, suggesting room for improvement in correctly classifying positive cases.

On the other hand, the values for the ‘tomography’ class are low. Sensitivity and precision are 0, meaning that the model did not correctly identify any positive cases. The F1 score is undefined, as the model did not correctly identify this class at all, indicating very poor performance for this term.

The values for the ‘xray’ class show mixed performance. Sensitivity is high, indicating that the model correctly identified all positive cases. Specificity is also high, showing that the model can correctly identify most negative cases. However, precision is relatively low, suggesting that many of the positive classifications may be false positives. The F1 score indicates a moderate balance between precision and sensitivity.

For the ‘angiogram’ class, the values show reasonable performance. Sensitivity indicates that the model correctly identified most positive cases. Specificity is high, indicating that most negative cases were correctly identified. However, precision is relatively low, suggesting that some of the positive classifications may be false positives. The F1 score indicates a moderate balance between precision and sensitivity.

For the ‘angiography’ class, sensitivity and precision are 0, indicating that the model did not correctly identify any positive cases. The F1 score is undefined, as the model did not correctly identify this class at all, indicating very poor performance for this term.

The ‘abdominal’ class also yielded sensitivity and precision of 0, indicating that the model did not correctly identify any positive cases. The F1 score is undefined, as the model did not correctly identify this class at all, indicating very poor performance for this term.

For the ‘contrast’ class, sensitivity is 0, indicating that the model did not correctly identify any positive cases. The F1 score is undefined, as the model did not correctly identify this class at all, indicating very poor performance for this term.

The model demonstrated promising performance in classifying different types of medical exams. However, certain classes presented greater difficulty in accurate classification, resulting in confusions with other classes. These findings offer valuable insights into the strengths and limitations of the proposed generative model for generating descriptive text for medical images.

An important observation is the interchangeability of some terms, such as ‘tomography’ being equivalent to ‘ct’. This indicates consistent results and generally good performance of the model in classifying medical exams.

The metrics of accuracy, sensitivity, specificity, precision, and negative predictive value showed relatively high values, indicating the model’s capability to make correct classifications for both positive and negative examples in most cases. This indicates the model’s ability to learn relevant patterns in medical images and generate accurate textual descriptions.

However, some sensitivity values equal to 0 indicate that the model failed to identify any positive cases for certain terms. Additionally, the presence of undefined values for precision and F1 score indicates limitations in interpreting these metrics due to the lack of positive predictions. The analysis of the metrics demonstrates varied performance for different terms, with some exhibiting positive values for sensitivity, specificity, precision, and F1 score, indicating good performance in correctly identifying positive and negative cases. However, other terms show inferior performance with low or undefined values for certain metrics.

It is important to acknowledge that variations in the results suggest the model’s performance may be influenced by different factors, such as data distribution or specific characteristics of each class. These variations are noticeable in metrics such as sensitivity, specificity, and precision.

Therefore, while the results are generally good, continuous refinement and optimization of the model are essential. Exploring different approaches and techniques, such as fine-tuning weights, data augmentation, and selection of relevant attributes, will be beneficial. Additionally, conducting further validations and testing on external datasets is crucial to verify the model’s generalization capabilities.

### 4.2. Confusion Matrix of Body Part

The confusion matrix presented in Table 15 represents the predictions of the best model concerning the body part of the medical images. Analyzing this confusion matrix allows us to draw several conclusions.

The classification of ‘chest’ shows generally good performance, with 39 cases correctly predicted. There are some confusions with ‘abdomen’ (three incorrect cases) and ’lung’ (ten incorrect cases). However, the majority of predictions for ‘chest’ are correct, indicating good performance in identifying this body part.

The model exhibits satisfactory performance in the classification of ‘abdomen’, with 47 cases correctly predicted. There are some confusions with ‘artery’ (two incorrect cases) and ’lung’ (two incorrect cases). Overall, the model can adequately identify ’abdomen’, but there are instances where confusions with other body parts occur.

In the classification of ‘artery’, the model shows relatively good performance, with 24 cases correctly predicted. However, there are confusions with ‘abdomen’, where 10 cases were incorrectly identified, and with ‘chest’, where a single incorrect prediction occurred. These confusions may be due to the anatomical proximity between arteries and these specific body regions, making it visually challenging for the model to distinguish between them. Therefore, it is important to continue refining the model to improve accuracy in these distinctions and reduce confusions between ‘artery’, ‘abdomen’, and ‘chest’.

In the classification of ‘lung’, with 16 cases correctly predicted, the model showed reasonable performance. However, some confusions occurred with ‘chest’ (12 incorrect cases) and ‘abdomen’ (six incorrect cases). These confusions can be attributed to the fact that the lungs are located in both the chest and abdominal regions. This anatomical overlap can make it visually challenging for the model to correctly distinguish between ‘lung’, ‘chest’, and ‘abdomen’. Therefore, it is necessary to enhance the model’s ability to accurately differentiate between these body parts, considering more specific characteristics related to the lungs in medical images.

The model encountered difficulties in the classification of ‘lobe’, with only one case correctly predicted. Most of the predictions for ‘lobe’ were confused with ‘abdomen’ (22 incorrect cases). These confusions may be attributed to the fact that ‘lobe’, in the dataset used, refers to the right and left lobes of the liver, which are located in the abdomen. This anatomical proximity between ‘lobe’ and ‘abdomen’ can make it challenging for the model to distinguish between them, especially considering the visual characteristics of medical images. Therefore, it is necessary to improve the model’s ability to accurately differentiate between the liver lobes (‘lobe’) and the abdomen (‘abdomen’) by considering more specific and discriminative features present in the medical images.

The model’s performance in the classification of ‘head’ shows that there is room for improvement, as only one case was correctly predicted. There were significant confusions with ‘abdomen’ (14 incorrect cases). These confusions may be justified by the fact that, in the context of the dataset used, ‘head’ refers to a specific part of the pancreas located in the abdomen. This association between ‘head’ and the pancreas, which is an abdominal structure, may lead to confusions in the classification, as the model may visually interpret the features present in the images similarly for both classes. Thus, anatomical proximity and visual overlap between ‘head’ and ‘abdomen’ can make it challenging for the model to distinguish between them accurately. Therefore, it is necessary to consider approaches that explore more specific features related to the pancreas, as well as more advanced image processing techniques, to improve the model’s ability to differentiate these body parts accurately.

The model obtained unsatisfactory performance in the classification of ‘liver’, correctly predicting none of the cases, and all cases were erroneously classified as ‘abdomen’. The confusions observed in the classification of ‘liver’ can be attributed to the fact that the liver is located in the abdomen. This anatomical and visual proximity between the liver and other abdominal structures can lead to confusions in the classification, as the visual features of these structures may overlap in medical images. The liver is a large and complex organ occupying a central position in the abdomen. Its shape, texture, and location can vary between individuals and in different medical images. This variation can make the precise segmentation and identification of the liver challenging, leading to confusions with other body parts present in the same anatomical region.

The ‘pelvi’ class presented consistent classification results, with all predictions correctly made by the model. This indicates satisfactory performance in identifying images corresponding to the pelvic region of the body. The fact that the model correctly classified all images associated with the ’pelvi’ class indicates its ability to recognize the relevant visual characteristics of this specific body region. This suggests that the distinctive features of pelvic medical images were well learned and captured by the model during training.

The ‘bone’ class showed moderate performance in the classification of medical images. The model was able to correctly identify six cases related to this class but with confusions in other categories, such as ‘abdomen’ and ‘chest’. The confusion observed between ’bone’ and other classes can be explained by the presence of similar or overlapping visual features in the images. For example, in abdominal (‘abdomen’) and chest (‘chest’) medical images, bones may be partially visible, resulting in confusions in the classification.

The ‘pulmonary’ class presented poor performance in the classification of medical images. The model was unable to correctly identify four cases related to this class but confused them with other categories, such as ‘chest’ and ‘lung’. As the ‘pulmonary’ class is associated with pulmonary structures such as the lungs, bronchi, and pulmonary blood vessels, the confusions observed with other classes may be attributed to the visual similarity between pulmonary structures and other adjacent structures in the chest, such as the chest (‘chest’) and lungs (‘lung’). This visual overlap can make it difficult for the model to distinguish precisely between the categories during classification.

The ‘aortic’ class presented poor performance in the classification of medical images. The model was unable to correctly identify any cases related to this class, presenting confusions with other categories, such as ‘abdomen’ and ‘artery’. The ‘aortic’ class is associated with the aorta, which is the largest artery in the human body and plays a crucial role in transporting blood to various parts of the body. The confusions observed with other classes can be explained by the visual similarity between the aorta and other structures, such as arteries in general (‘artery’) and the abdominal region (‘abdomen’). This visual overlap and shared characteristics can make it difficult for the model to distinguish precisely between the categories during classification.

Regarding the ‘vein’ class, the model showed poor performance, with no cases correctly predicted. However, there were some confusions with other classes, especially with the ‘abdomen’ class. The confusions may be attributed to the fact that the vein data in the dataset are related to the liver, which is located in the abdominal region. This anatomical and visual proximity between hepatic veins and other abdominal structures can make it challenging to distinguish precisely during the classification process.

The ’hip’ class represents the hip region in medical images. The model presented unsatisfactory results in the classification of the hip, with only one case correctly predicted. However, there were some confusions with other classes, such as ‘pelvi’ and ‘bone’. These confusions can be justified by the fact that the hip and pelvis are closely related anatomically, and the precise distinction between these structures can be challenging. Additionally, the hip is composed of several bone structures, such as the hip bone (ilium, ischium, and pubis) and the femur, which may overlap with other nearby bone regions.

The ’neck’ class represents the neck region in medical images. The model obtained poor results in the classification of this class, with no cases correctly predicted, and there were confusions with other classes, such as ‘pelvi’ and ‘bone’, and ‘chest’. The confusion with ‘chest’ can be attributed to the transition between the neck and abdomen, which may not be clearly distinct, especially when dealing with images with structural overlap. The confusion with ‘pelvi’ and ‘bone’ can be due to the presence of cervical bones in the neck, such as cervical vertebrae. These bone structures may be similar to those found in other parts of the body, making it challenging for the model to classify the neck accurately.

The ’femoral’ class represents the femoral artery region in medical images. Although the model achieved some successes in the classification of this class, there were significant confusions with the ‘bone’ and ‘pelvi’ classes. The confusion with the ‘bone’ class can be attributed to the femur being a bone, which makes it challenging to precisely distinguish between the two classes. The confusion with the ‘pelvi’ class can be explained by the close location of the femoral artery to the pelvic region. In some images, the femoral artery may be partially within the pelvic region, leading to confusions between the two classes.

In Table 16, the metrics of the confusion matrix for the best model regarding the body part of the images are presented. This table allows us to evaluate the performance of the best model and draw conclusions.

Upon analyzing the metrics of the confusion matrix for all provided classes, it is evident that considerable variation exists in the results. Let us discuss each class in a general context and evaluate whether the values are considered good or poor.

The metrics for the ‘chest’ class indicate relatively good performance. The true positive rate (TPR) of 0.72 and the true negative rate (TNR) of 0.92 demonstrate a satisfactory rate of correct identification for both positive and negative cases. The positive predictive value (PPV) of 0.68 indicates that about 68% of cases classified as positive are indeed positive. The accuracy (ACC) of 0.88 is also quite reasonable. The F1 score of 0.70, which is the harmonic mean between precision and recall, indicates a balanced performance between these metrics.

Conversely, the results for the ‘abdomen’ class are less favorable. Although the TPR of 0.90 is high, suggesting a good rate of correct identification for positive cases, the TNR of 0.66 indicates a relatively low rate of correct identification for negative cases. The PPV of 0.38 indicates that only 38% of cases classified as positive are indeed positive. The accuracy (ACC) of 0.71 is also relatively low. The F1 score of 0.53 shows moderate performance, but there is still room for improvement.

In contrast, the metrics for the ‘artery’ class are generally positive. The TPR of 0.69 indicates a reasonable rate of correct identification for positive cases, while the TNR of 0.97 suggests a high rate of correct identification for negative cases. The PPV of 0.77 is relatively high, indicating that 77% of cases classified as positive are indeed positive. The accuracy (ACC) of 0.94 is also quite satisfactory. The F1 score of 0.73 demonstrates a good balance between precision and recall.

As for the ‘lung’ class, the results are not as positive. The TPR of 0.44 indicates a relatively low rate of correct identification for positive cases, while the TNR of 0.92 suggests a high rate of correct identification for negative cases. The PPV of 0.48 indicates that only 48% of cases classified as positive are indeed positive. The accuracy (ACC) of 0.85 is also moderate. The F1 score of 0.46 shows performance below the ideal in terms of balance between precision and recall.

The metrics for the ‘lobe’ class reveal very low performance. The TPR of 0.04 indicates an extremely low rate of correct identification for positive cases. On the other hand, the TNR of 0.99 suggests a high rate of correct identification for negative cases. The PPV of 0.33 indicates that only 33% of cases classified as positive are indeed positive. The accuracy (ACC) of 0.91 is moderate, but the F1 score of 0.07 indicates very low performance in terms of balance between precision and recall.

The metrics for the ‘head’ class also do not present satisfactory performance. The TPR of 0.07 indicates a very low rate of correct identification for positive cases, while the TNR of 1.00 suggests a high rate of correct identification for negative cases. The PPV of 0.50 indicates that only 50% of cases classified as positive are indeed positive. The accuracy (ACC) of 0.95 is reasonable, but the F1 score of 0.12 indicates low performance in terms of balance between precision and recall.

Regarding the ‘liver’ class, there are no positive cases (TP = 0), which prevents the calculation of PPV and F1 score. However, the TNR of 1.00 suggests a high rate of correct identification for negative cases, and the negative predictive value (NPV) of 0.96 indicates a high proportion of correctly classified negative cases. The accuracy (ACC) of 0.96 is also considered good.

The metrics for the ‘pelvi’ class reveal mixed performance. The TPR of 1.00 indicates a high rate of correct identification for positive cases, while the TNR of 0.96 suggests a high rate of correct identification for negative cases. However, the PPV of 0.15 indicates that only 15% of cases classified as positive are indeed positive. The NPV of 1.00 indicates that all negative cases were correctly identified. The accuracy (ACC) of 0.96 is reasonable, and the F1 score of 0.27 shows moderate performance in terms of balance between precision and recall.

The metrics for the ‘bone’ class indicate moderate performance. The TPR of 0.55 suggests a reasonable rate of correct identification for positive cases, while the TNR of 0.96 indicates a high rate of correct identification for negative cases. The PPV of 0.33 indicates that only 33% of cases classified as positive are indeed positive. The NPV of 0.98 indicates a high proportion of correctly classified negative cases. The accuracy (ACC) of 0.94 is considered good. The F1 score of 0.41 shows moderate performance in terms of balance between precision and recall.

Regarding the ‘pulmonary’ class, there are no positive cases (TP = 0), which prevents the calculation of PPV and F1 score. However, the TNR of 1.00 suggests a high rate of correct identification for negative cases. The NPV of 0.98 indicates a high proportion of correctly classified negative cases. The accuracy (ACC) of 0.98 is considered good.

Similarly to the ‘pulmonary’ class, there are no positive cases (TP = 0) for the ‘aortic’ class, which prevents the calculation of PPV and F1 score. The TNR of 1.00 suggests a high rate of correct identification for negative cases. The NPV of 0.98 indicates a high proportion of correctly classified negative cases. The accuracy (ACC) of 0.98 is considered good.

As with the previous classes, there are no positive cases (TP = 0) for the ‘vein’ class, preventing the calculation of PPV and F1 score. The TNR of 1.00 suggests a high rate of correct identification for negative cases. The NPV of 0.96 indicates a high proportion of correctly classified negative cases. The accuracy (ACC) of 0.96 is considered reasonable.

The metrics for the ‘hip’ class indicate relatively good performance. The TPR of 0.25 suggests a moderate rate of correct identification for positive cases, while the TNR of 1.00 suggests a high rate of correct identification for negative cases. The PPV of 1.00 indicates that all cases classified as positive are indeed positive. The NPV of 0.99 indicates a high proportion of correctly classified negative cases. The accuracy (ACC) of 0.99 is considered good. The F1 score of 0.40 shows moderate performance in terms of balance between precision and recall.

Regarding the ‘neck’ class, there are no positive cases (TP = 0), which prevents the calculation of PPV and F1 score. The TNR of 1.00 suggests a high rate of correct identification for negative cases. The NPV of 0.96 indicates a high proportion of correctly classified negative cases. The accuracy (ACC) of 0.96 is considered reasonable.

The metrics for the ‘femoral’ class indicate moderate performance. The TPR of 0.14 suggests a relatively low rate of correct identification for positive cases, while the TNR of 1.00 indicates a high rate of correct identification for negative cases. The PPV of 1.00 indicates that all cases classified as positive are indeed positive. The NPV of 0.98 indicates a high proportion of correctly classified negative cases. The accuracy (ACC) of 0.98 is considered good. The F1 score of 0.25 shows moderate performance in terms of balance between precision and recall.

In summary, the analysis of the confusion matrix for body parts in medical images reveals a moderate performance of the model in classifying the body parts. While some classes show relatively good performance, such as ‘chest,’ ‘abdomen,’ and ‘artery,’ indicating the model’s ability to identify vascular structures in medical images, other classes demonstrate lower performance, e.g., ‘lobe,’ ‘head,’ and ‘liver,’ which can be attributed to anatomical proximity or a lack of clear visual distinction between these structures.

Despite the confusions, it is encouraging that the model could capture the general context and provide relevant information about the body parts present in the medical images. This suggests that the model has the potential to assist in description and classification tasks, even if it requires refinement to improve accuracy in choosing specific words.

However, it is essential to emphasize that the model’s performance can be enhanced through more advanced approaches, such as using sophisticated image processing techniques, incorporating additional contextual information, and training on more diverse and balanced datasets.

Therefore, while the model shows a moderate performance in classifying body parts in medical images, there is room for improvements and refinements that can enable a more precise and specific description of the anatomical structures present in the images.

In summary, the analysis of the metrics from the confusion matrix reveals varied results for different classes. Some classes exhibit satisfactory performance, with high rates of correct identification of positive and negative cases, as well as positive values for precision, recall, and F1 score metrics. However, other classes show lower performance, with low rates of correct identification of positive cases, low values for precision and recall metrics, and low F1 scores.

Additionally, some classes had no positive cases in the sample, making it challenging to properly evaluate metrics such as precision and F1 score for these cases. Nevertheless, these classes demonstrated a high rate of correct identification for negative cases and high values of negative predictive value (NPV), indicating that negative cases were correctly classified.

### 4.3. Confusion Matrix of the Identified Problem

The presented confusion matrix in Table 17 illustrates the predictions made by the best model regarding the identified problem in the images. By analyzing this confusion matrix, several conclusions can be drawn.

In the ‘nodule’ class, the model demonstrated relatively good performance, correctly identifying 30 cases. Although some confusion occurred with other classes, such as ‘hepatic’, ‘aneurysm’, ‘tumor’, and ‘cystic’, these can be attributed to the complex nature of nodules, which can vary in size, shape, and location. Nevertheless, it is crucial to highlight that even with these confusions, the model made several correct predictions for nodule cases, indicating its ability to capture distinctive characteristics of nodules despite facing challenges with more complex cases.

Conversely, the ‘hepatic’ class exhibited poor performance in terms of classification. While the model made some correct predictions, significant confusions arose with other classes, such as ‘cystic’ and ‘pancreatic’. This confusion can be explained by the visual overlap between the liver and pancreas in medical images, leading to incorrect predictions due to overlapping features, making precise differentiation challenging for the model.

The ‘aneurysm’ class displayed reasonable performance in terms of classification, with 15 correct predictions. However, there were also some confusions with other classes, such as ‘pancreatic’ and ‘cystic’. These confusions can be attributed to certain types of cysts exhibiting similar characteristics to aneurysms, making the distinction between these classes challenging. Despite these confusions, the model was able to correctly identify a significant portion of aneurysm cases.

Regarding the ‘fracture’ class, the model performed well, correctly identifying 30 cases with minimal confusion with other classes.

Similarly, the ‘normal’ class also showed good performance, with 21 cases correctly identified and minimal confusion with other classes.

However, the ‘tumor’ class had moderate performance in terms of classification. The model made a significant number of correct predictions for tumors. Nevertheless, there were some confusions with other classes, such as ‘cystic’ and ‘pancreatic’. These confusions can be attributed to certain types of tumors having visual similarities with hepatic lesions and cysts, making the precise differentiation between these classes challenging. Despite the confusions, the model successfully identified nine cases of tumors, indicating its ability to capture certain distinctive characteristics of this condition.

For the ‘pancreatic’ class, the model displayed relatively good performance, correctly identifying 15 cases. However, there were some confusions with other classes, such as ‘hepatic’, ‘cystic’, and ‘hematoma’. This confusion can be explained, in part, by the visual overlap between the liver and pancreas in medical images, leading to incorrect predictions due to overlapping features, making precise differentiation challenging for the model.

In the ‘effusion’ class, there were only confusions with other classes, such as ‘nodule’, ‘cystic’, and ‘tumor’. These confusions can be justified by the presence of overlapping or similar visual characteristics among these conditions. For example, some effusions may exhibit mass or lesion-like characteristics resembling nodules or cysts. Likewise, some effusions may appear visually similar to tumors in terms of their appearance in medical images.

The ‘cystic’ class showed moderate performance in terms of classification, with nine correct predictions. However, some confusions occurred with other classes, such as ‘pancreatic’. These confusions can be attributed to the presence of shared or overlapping visual characteristics between these conditions. For example, some cysts may exhibit characteristics of nodules or hepatic lesions, making the distinction between these classes challenging. Likewise, the presence of cysts in organs such as the pancreas may visually resemble other conditions, such as pancreatic tumors.

The ‘hematoma’ class displayed poor performance, with all predictions being incorrect. There was a confusion with ‘cystic’, which can be justified by the fact that some hematomas may appear similar to cysts or nodules, especially when there is fluid or blood accumulation in a specific region. Likewise, the presence of hematomas in hepatic organs may resemble hepatic lesions, justifying the confusion with the ’pancreatic’ class, making the differentiation between the classes challenging.

The ‘cyst’ class generally exhibited good performance in terms of classification, with the majority of predictions being correct, considering that the terms ‘cyst’ and ‘cystic’ are interchangeable. However, some confusion occurred with the ‘pancreatic’ class. These confusions can be justified by the fact that some cysts may share characteristics with hepatic lesions, making the distinction between these classes challenging. Despite the confusion, the model was able to correctly identify most cases of cysts, indicating its ability to capture distinctive characteristics of this condition.

In Table 18, we present the metrics of the confusion matrix for the best model regarding the identified problem in the images. This table allows for the evaluation of the performance of the best model and drawing relevant conclusions.

Upon analyzing the metrics of the confusion matrix for all provided classes, we observe considerable variation in the results. Let us discuss each class in a general context and analyze whether the values are considered good or poor.

The class ‘nodule’ exhibits a reasonable performance. The true positive rate (TPR) of 0.59 indicates a moderate rate of correct identification of positive cases, while the true negative rate (TNR) of 0.95 suggests a high rate of correct identification of negative cases. The positive predictive value (PPV) of 0.71 indicates that approximately 71% of cases classified as positive are truly positive. The negative predictive value (NPV) of 0.91 indicates a high proportion of correctly classified negative cases. The accuracy (ACC) of 0.88 is considered good. The F1 score of 0.65 shows moderate performance in terms of the balance between precision and recall.

On the other hand, the class ‘hepatic’ presents poor performance. The TPR of 0.11 indicates a low rate of correct identification of positive cases, while the TNR of 0.97 suggests a high rate of correct identification of negative cases. The PPV of 0.38 indicates that only about 38% of cases classified as positive are truly positive. The NPV of 0.85 indicates a reasonable proportion of correctly classified negative cases. The accuracy (ACC) of 0.83 is considered fair. The low F1 score of 0.17 indicates poor performance in terms of the balance between precision and recall.

Regarding the class ‘aneurysm,’ it shows reasonable performance. The TPR of 0.50 indicates a moderate rate of correct identification of positive cases, while the TNR of 0.94 suggests a high rate of correct identification of negative cases. The PPV of 0.50 indicates that approximately 50% of cases classified as positive are truly positive. The NPV of 0.94 indicates a high proportion of correctly classified negative cases. The accuracy (ACC) of 0.89 is considered good. The F1 score of 0.50 shows moderate performance in terms of the balance between precision and recall.

Moving on to the class ‘fracture,’ it presents generally good performance. The TPR of 0.88 indicates a high rate of correct identification of positive cases, while the TNR of 0.99 suggests a very high rate of correct identification of negative cases. The PPV of 0.91 indicates that about 91% of cases classified as positive are truly positive. The NPV of 0.98 indicates a high proportion of correctly classified negative cases. The accuracy (ACC) of 0.98 is considered good. The F1 score of 0.90 shows solid performance in terms of the balance between precision and recall.

As for the class ‘normal,’ it shows reasonable performance. The TPR of 0.66 indicates a moderate rate of correct identification of positive cases, while the TNR of 0.98 suggests a high rate of correct identification of negative cases. The PPV of 0.84 indicates that approximately 84% of cases classified as positive are truly positive. The NPV of 0.96 indicates a high proportion of correctly classified negative cases. The accuracy (ACC) of 0.95 is considered good. The F1 score of 0.74 shows moderate performance in terms of the balance between precision and recall.

Similarly, the class ‘tumor’ shows reasonable performance. The TPR of 0.50 indicates a moderate rate of correct identification of positive cases, while the TNR of 0.97 suggests a high rate of correct identification of negative cases. The PPV of 0.50 indicates that approximately 50% of cases classified as positive are truly positive. The NPV of 0.97 indicates a high proportion of correctly classified negative cases. The accuracy (ACC) of 0.94 is considered good. The F1 score of 0.50 shows moderate performance in terms of the balance between precision and recall.

However, the class ‘pancreatic’ presents poor performance. The TPR of 0.68 indicates a moderate rate of correct identification of positive cases, while the TNR of 0.88 suggests a relatively low rate of correct identification of negative cases. The PPV of 0.32 indicates that only about 32% of cases classified as positive are truly positive. The NPV of 0.97 indicates a high proportion of correctly classified negative cases. The accuracy (ACC) of 0.86 is considered fair. The F1 score of 0.43 shows moderate performance in terms of the balance between precision and recall.

Regarding the classes ‘effusion’, ‘hematoma’, and ‘cyst’, they have a number of positive cases equal to zero, making it difficult to properly evaluate precision, recall, and F1 score metrics. However, the TNR values for these classes are relatively high, indicating a high rate of correct identification of negative cases.

The model exhibited varied results in the classification of different classes of identified issues in medical images. For some classes, such as ‘nodule,’ ‘aneurysm,’ ‘fracture,’ and ‘normal,’ the performance was relatively good, with a significant number of correct predictions. However, there were confusions with other classes, indicating visual feature overlap between different medical conditions.

On the other hand, some classes, such as ‘hepatic,’ ‘effusion,’ and ‘hematoma,’ presented more challenging performance. There were considerable confusions with other classes, indicating visual feature overlap and difficulty in distinguishing these conditions.

Regarding the confusion matrix metrics, some classes showed reasonable performance, such as ‘nodule,’ ‘aneurysm,’ ‘fracture,’ and ‘normal,’ with moderate rates of correct identification of positive cases, high rates of correct identification of negative cases, and moderate F1 scores. Notably, the ‘fracture’ class exhibited generally good performance, with high rates of correct identification of positive cases, high rates of correct identification of negative cases, and a solid F1 score. On the other hand, the ‘hematoma’ and ‘cyst’ classes have a number of positive cases equal to zero, making it challenging to evaluate some metrics properly.

Conversely, the classes ‘hepatic,’ ‘effusion,’ and ‘hematoma’ exhibited inferior performance, with low rates of correct identification of positive cases and/or low F1 scores.

To enhance classification in these classes, more advanced approaches can be employed, such as specific image processing techniques and a more comprehensive dataset that includes a broader range of examples.

## 5. Conclusions

The automatic generation of descriptions for medical images has emerged as a promising research area, holding significant potential to support healthcare professionals in the interpretation and analysis of clinical exams. The implementation of such solutions can yield substantial benefits for both patients and professionals by streamlining decision-making and enhancing the quality of medical care. In this study, we undertook the development and evaluation of a versatile generative model for medical image descriptions using the ROCO dataset.

Our work explores the generalization of description generation models for different medical image modalities, such as X-rays, magnetic resonance imaging, or computed tomography, and for various medical conditions. This approach broadens the applicability of the proposed models, allowing them to be employed in diverse clinical scenarios and providing greater flexibility in generating descriptions for medical images.

An important aspect to consider is that having a more generalist model reduces the complexity of the generated descriptions and may lead to an increase in confusion, especially when describing specific nuances of certain image modalities or medical conditions. Therefore, it is crucial to strike a balance between generalization and the ability to capture detailed and relevant features of medical images with the aim of assisting physicians and healthcare professionals in diagnoses and decision-making.

To achieve our objectives, we explored the applicability of various models, including MobileNetV2, DenseNet201, ResNet152V2, NASNetLarge, VGG19, Xception, InceptionV3, and InceptionResNetV2, for generating medical descriptions, with the aim of achieving generalization across diverse image modalities and medical conditions.

By utilizing a variety of feature extraction models and the transformer approach, this study provides a more comprehensive analysis and enables the identification of which models may be more effective in generating descriptions for medical images.

Although promising results were obtained in generating descriptions for medical images, the quality of the generated descriptions still exhibited limitations. Some descriptions contained semantic errors or lacked relevant specific details. These limitations could be attributed, in part, to the availability and representativeness of the ROCO data, which can influence the obtained results. Additionally, the techniques used for generating descriptions may not fully capture the complexity of medical images and their clinical interpretation. It is crucial to consider these limitations when interpreting and generalizing the results of this study.

As recommendations for future research, we emphasize the importance of further exploring the influence of different techniques and approaches. This may involve the use of more advanced neural network architectures, the implementation of interpretability in generative models, and the utilization of even larger and more diversified datasets. Additionally, seeking a more appropriate metric for evaluating the quality of descriptions and considering clinical validation can contribute to a deeper comparative analysis between generated descriptions and those provided by experts.

## Figures and Tables

**Figure 1 bioengineering-10-01098-f001:**
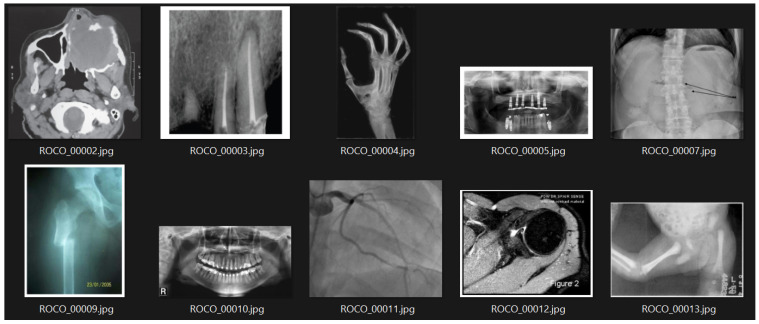
Images belonging to the ’radiology’ subset of the ROCO dataset.

**Figure 2 bioengineering-10-01098-f002:**
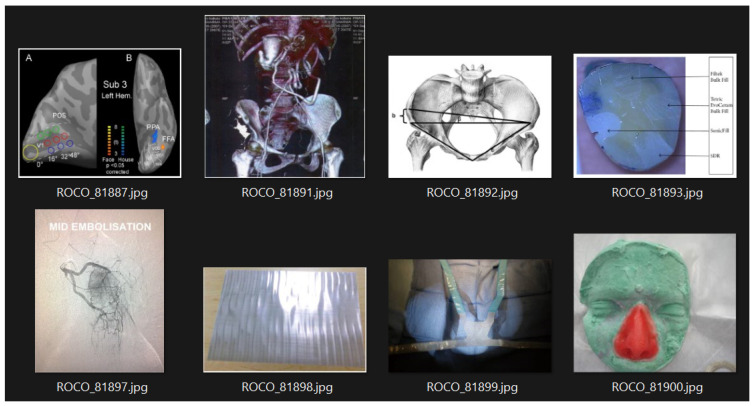
Images belonging to the ’non-radiology’ subset of the ROCO dataset.

**Figure 3 bioengineering-10-01098-f003:**
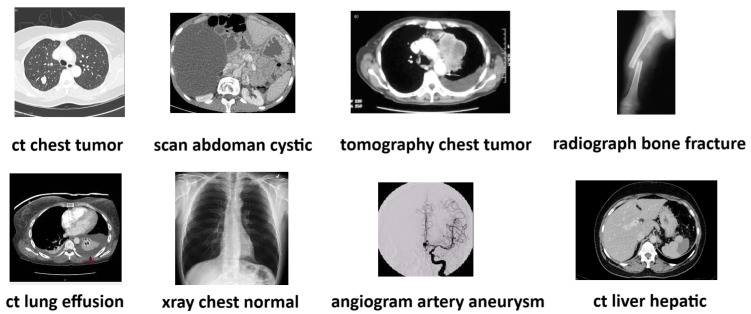
Example of random images from the dataset and their respective three-word descriptions.

**Figure 4 bioengineering-10-01098-f004:**
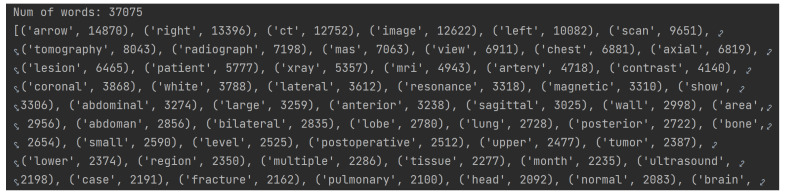
Listing of a portion of the frequency of all keywords.

**Figure 5 bioengineering-10-01098-f005:**
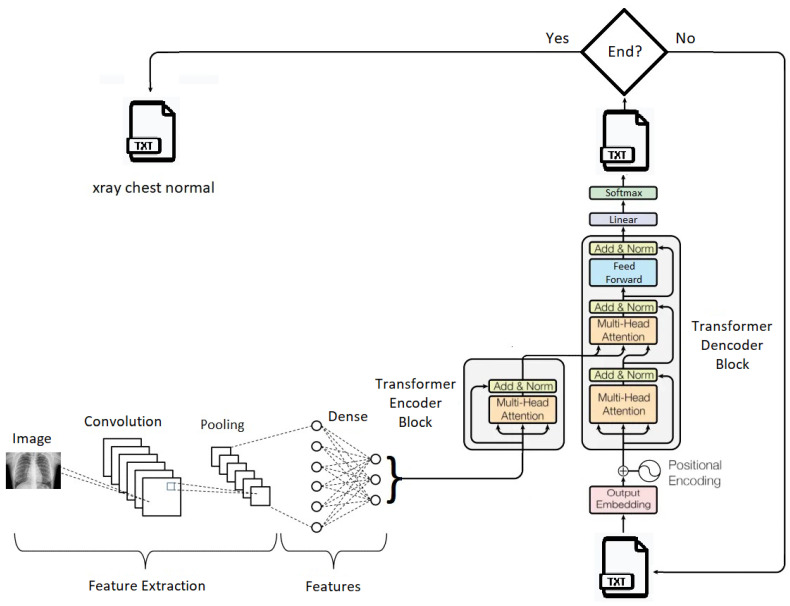
The proposed generative model.

**Figure 6 bioengineering-10-01098-f006:**
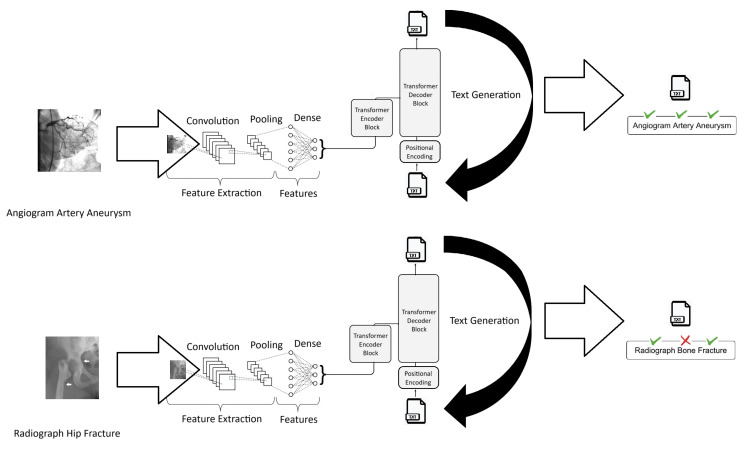
Examples of input images and true and generated descriptions.

**Table 1 bioengineering-10-01098-t001:** Keywords of interest for each category.

Type of Examination	Body Part	Identified Problem
ct (12,752)	chest (6881)	tumor (2387)
scan (9651)	artery (4718)	fracture (2162)
tomography (8043)	abdomen (2856)	normal (2083)
radiograph (7198)	lobe (2780)	cystic (1700)
xray (5357)	lung (2728)	cyst (1477)
mri (4943)	bone (2654)	effusion (1371)
contrast (4140)	tissue (2277)	calcification (1179)
resonance (3318)	mouth (2235)	nodule (1158)
magnetic (3310)	pulmonary (2100)	hepatic (997)
abdominal (3274)	head (2092)	node (981)
ultrasound (2198)	brain (2038)	stent (901)
angiography (1680)	vein (1897)	heterogeneous (889)
angiogram (1220)	liver (1695)	pancreatic (874)
catheter (1020)	ventricle (1675)	weak (858)
	pelvi (1635)	aneurysm (842)
	kidney (1468)	edema (833)
	spine (1321)	irregular (752)
	neck (1307)	dilatation (746)
	pleural (1294)	absces (745)
	muscle (1293)	disease (743)
	renal (1266)	apical (700)
	coronary (1258)	hematoma (688)
	femoral (1252)	fistula (608)
	cervical (1196)	cancer (505)
	atrium (1192)	
	bowel (1171)	
	aorta (1098)	
	aortic (1064)	
	hip (1018)	
	heart (620)	
	tooth (619)	

**Table 2 bioengineering-10-01098-t002:** Selected groups of keywords of interest.

Type of Examination	Body Part	Identified Problem	Quantity of Images
ct	chest	tumor	52
ct	artery	aneurysm	63
ct	abdomen	tumor	38
ct	abdomen	cystic	58
ct	abdomen	hepatic	47
ct	abdomen	pancreatic	35
ct	abdomen	hematoma	38
ct	lobe	hepatic	51
ct	lung	effusion	56
ct	head	pancreatic	56
ct	vein	hepatic	34
ct	liver	cyst	35
ct	liver	hepatic	28
scan	chest	nodule	77
scan	abdomen	cystic	23
scan	lobe	hepatic	18
scan	lung	effusion	18
scan	lung	nodule	41
scan	pulmonary	nodule	17
scan	head	pancreatic	13
tomography	chest	tumor	29
tomography	lobe	nodule	41
tomography	lobe	hepatic	21
tomography	lung	effusion	20
tomography	lung	nodule	34
tomography	pulmonary	nodule	21
radiograph	chest	normal	47
radiograph	bone	fracture	41
radiograph	pelvi	fracture	44
radiograph	neck	fracture	31
radiograph	femoral	fracture	31
radiograph	hip	fracture	32
xray	chest	normal	65
contrast	artery	aneurysm	32
abdominal	aortic	aneurysm	35
angiography	artery	normal	39
angiogram	artery	aneurysm	58
		Total	1419

**Table 3 bioengineering-10-01098-t003:** Mean and standard deviation of accuracy.

CNN	μ	σ	Min	Max	μ−σ	μ+σ
DenseNet201	0.7365	0.0114	0.7151	0.7610	0.7250	0.7479
ResNet152V2	0.7154	0.0138	0.6868	0.7359	0.7016	0.7292
NASNetLarge	0.7050	0.0085	0.6891	0.7210	0.6965	0.7135
VGG19	0.7431	0.0167	0.7056	0.7628	0.7264	0.7598
Xception	0.7313	0.0116	0.7069	0.7499	0.7197	0.7429
InceptionV3	0.7236	0.0124	0.6992	0.7472	0.7111	0.7360
InceptionResNetV2	0.6985	0.0089	0.6823	0.7165	0.6896	0.7074
MobileNetV2	0.7056	0.0225	0.6603	0.7345	0.6831	0.7281
Global	0.7199	0.0206	0.6603	0.7628	0.6992	0.7404

**Table 4 bioengineering-10-01098-t004:** Results of the DenseNet201 family.

ID	OPTZ	TL	TR	IA	Loss	ACC	Epochs	Training Time (s)	Test Time (s)
1	Adam		√		0.7487	0.7276	20	6520	246
2	AdamW		√		0.7727	0.7205	6	1656	250
3	Adadelta		√		0.7601	0.7253	8	2185	248
4	Adafactor		√		0.7539	0.7275	7	1896	259
5	Adam		√	√	0.7894	0.7151	20	5662	249
6	AdamW		√	√	0.7461	0.7271	6	1706	281
7	Adadelta		√	√	0.7390	0.7279	6	1784	293
8	Adafactor		√	√	0.7199	0.7356	6	1812	314
9	Adam	√			0.7438	0.7443	10	2887	293
10	AdamW	√			0.7244	0.7406	4	1160	298
11	Adadelta	√			0.7223	0.7413	8	2345	306
12	Adafactor	√			0.7190	0.7446	5	1476	319
13	Adam	√		√	0.6907	0.7413	18	5463	334
14	AdamW	√		√	0.6890	0.7483	4	1454	400
15	Adadelta	√		√	0.6818	0.7549	10	3400	357
16	Adafactor	√		√	0.6807	0.7532	7	3010	443
17	Adam	√	√		0.7519	0.7337	11	4778	345
18	AdamW	√	√		0.7534	0.7439	4	1203	329
19	Adadelta	√	√		0.7408	0.7449	5	1540	346
20	Adafactor	√	√		0.7190	**0.7610**	6	2205	420
21	Adam	√	√	√	0.7236	0.7262	18	6640	424
22	AdamW	√	√	√	0.7349	0.7242	4	1530	436
23	Adadelta	√	√	√	0.7059	0.7343	8	2733	366
24	Adafactor	√	√	√	0.7044	0.7318	4	1649	433
							Total	66,694	7989

**Table 5 bioengineering-10-01098-t005:** Results of the ResNet152V2 family.

ID	OPTZ	TL	TR	IA	Loss	ACC	Epochs	Training Time (s)	Test Time (s)
25	Adam		√		0.8128	0.7020	20	7844	293
26	AdamW		√		0.8027	0.7113	4	1654	385
27	Adadelta		√		0.7894	0.7174	10	3999	294
28	Adafactor		√		0.7840	0.7127	7	2815	308
29	Adam		√	√	0.8791	0.6868	11	4493	299
30	AdamW		√	√	0.8525	0.6929	5	2059	332
31	Adadelta		√	√	0.8480	0.6908	10	4116	355
32	Adafactor		√	√	0.8499	0.6919	6	2428	337
33	Adam	√			0.7817	0.7277	15	5974	338
34	AdamW	√			0.7539	0.7278	11	4395	318
35	Adadelta	√			0.7524	0.7263	7	2912	366
36	Adafactor	√			0.7413	**0.7359**	9	3771	452
37	Adam	√		√	0.7865	0.7151	18	7519	382
38	AdamW	√		√	0.7712	0.7248	6	2548	372
39	Adadelta	√		√	0.7722	0.7228	7	2957	361
40	Adafactor	√		√	0.7533	0.7311	11	4672	384
41	Adam	√	√		0.7424	0.7237	19	7932	381
42	AdamW	√	√		0.7593	0.7124	4	1647	334
43	Adadelta	√	√		0.7469	0.7136	7	2891	359
44	Adafactor	√	√		0.7271	0.7246	8	3645	1042
45	Adam	√	√	√	0.7815	0.7033	20	9948	396
46	AdamW	√	√	√	0.7629	0.7214	4	1648	339
47	Adadelta	√	√	√	0.7523	0.7234	6	2713	453
48	Adafactor	√	√	√	0.7425	0.7298	11	5785	489
							Total	100,363	9371

**Table 6 bioengineering-10-01098-t006:** Results of the NASNetLarge family.

ID	OPTZ	TL	TR	IA	Loss	ACC	Epochs	Training Time (s)	Test Time (s)
49	Adam		√		0.8557	0.6914	16	7587	466
50	AdamW		√		0.8530	0.6980	5	2427	479
51	Adadelta		√		0.8526	0.6900	5	2418	495
52	Adafactor		√		0.8356	0.7011	8	3887	482
53	Adam		√	√	0.8414	0.7001	18	8774	490
54	AdamW		√	√	0.8509	0.6891	4	2005	551
55	Adadelta		√	√	0.8475	0.6969	8	3699	537
56	Adafactor		√	√	0.8459	0.6949	5	2494	523
57	Adam	√			0.8125	0.7084	11	5433	531
58	AdamW	√			0.8115	0.7095	4	1967	537
59	Adadelta	√			0.7938	0.7105	7	3798	674
60	Adafactor	√			0.7849	0.7173	6	3864	765
61	Adam	√		√	0.8122	0.7065	15	10,067	822
62	AdamW	√		√	0.7857	0.7089	6	3495	707
63	Adadelta	√		√	0.7780	0.7137	6	4172	832
64	Adafactor	√		√	0.7734	0.7081	5	3401	862
65	Adam	√	√		0.7838	0.7061	12	7191	768
66	AdamW	√	√		0.8015	0.7141	7	3520	572
67	Adadelta	√	√		0.8002	0.7121	4	2210	603
68	Adafactor	√	√		0.7813	**0.7210**	7	4260	768
69	Adam	√	√	√	0.8082	0.7021	11	6600	691
70	AdamW	√	√	√	0.8155	0.7010	5	2702	629
71	Adadelta	√	√	√	0.7998	0.7058	6	3095	609
72	Adafactor	√	√	√	0.7875	0.7134	10	6359	821
							Total	105,426	15,216

**Table 7 bioengineering-10-01098-t007:** Results of the VGG19 family.

ID	OPTZ	TL	TR	IA	Loss	ACC	Epochs	Training Time (s)	Test Time (s)
73	Adam		√		0.7734	0.7056	16	9480	310
74	AdamW		√		0.7534	0.7166	7	3384	158
75	Adadelta		√		0.7375	0.7252	11	5194	150
76	Adafactor		√		0.7231	0.7242	6	2831	150
77	Adam		√	√	0.8118	0.7126	12	5693	150
78	AdamW		√	√	0.7517	0.7258	16	7615	152
79	Adadelta		√	√	0.7382	0.7318	5	2336	150
80	Adafactor		√	√	0.7266	0.7273	6	2805	151
81	Adam	√			0.6648	0.7510	10	4685	152
82	AdamW	√			0.6530	0.7485	4	1887	158
83	Adadelta	√			0.6530	0.7483	8	3879	156
84	Adafactor	√			0.6485	0.7503	5	2459	166
85	Adam	√		√	0.6276	**0.7628**	18	8930	163
86	AdamW	√		√	0.6509	0.7456	4	2014	172
87	Adadelta	√		√	0.6325	0.7525	10	5040	171
88	Adafactor	√		√	0.6158	0.7607	14	7091	176
89	Adam	√	√		0.6659	0.7558	11	5467	168
90	AdamW	√	√		0.6608	0.7580	7	3410	240
91	Adadelta	√	√		0.6787	0.7538	4	3495	153
92	Adafactor	√	√		0.6784	0.7477	7	3179	163
93	Adam	√	√	√	0.6608	0.7586	10	4369	150
94	AdamW	√	√	√	0.6380	0.7552	13	6118	170
95	Adadelta	√	√	√	0.6307	0.7564	10	6210	174
96	Adafactor	√	√	√	0.6224	0.7593	5	2512	177
							Total	110,084	4080

**Table 8 bioengineering-10-01098-t008:** Results of the Xception family.

ID	OPTZ	TL	TR	IA	Loss	ACC	Epochs	Training Time (s)	Test Time (s)
97	Adam		√		0.7686	0.7215	16	2994	120
98	AdamW		√		0.7551	0.7223	9	1724	121
99	Adadelta		√		0.7416	0.7334	9	1728	118
100	Adafactor		√		0.7381	0.7289	7	1358	120
101	Adam		√	√	0.7961	0.7069	16	3174	121
102	AdamW		√	√	0.7709	0.7104	6	1244	134
103	Adadelta		√	√	0.7614	0.7154	6	1200	124
104	Adafactor		√	√	0.7687	0.7131	4	799	121
105	Adam	√			0.7521	0.7235	12	2342	124
106	AdamW	√			0.7493	0.7298	4	821	134
107	Adadelta	√			0.7416	0.7310	4	804	131
108	Adafactor	√			0.7381	0.7313	6	1208	132
109	Adam	√		√	0.7825	0.7306	20	4983	171
110	AdamW	√		√	0.7013	0.7449	4	870	139
111	Adadelta	√		√	0.6895	**0.7499**	6	1238	133
112	Adafactor	√		√	0.6848	0.7479	4	826	132
113	Adam	√	√		0.7112	0.7400	12	2371	133
114	AdamW	√	√		0.7237	0.7362	4	1003	124
115	Adadelta	√	√		0.7209	0.7354	4	1232	278
116	Adafactor	√	√		0.7134	0.7366	4	1483	274
117	Adam	√	√	√	0.6928	0.7359	20	3613	125
118	AdamW	√	√	√	0.6896	0.7413	4	772	141
119	Adadelta	√	√	√	0.6958	0.7393	6	1186	133
120	Adafactor	√	√	√	0.6816	0.7463	8	1599	142
							Total	40,571	3426

**Table 9 bioengineering-10-01098-t009:** Results of the InceptionV3 family.

ID	OPTZ	TL	TR	IA	Loss	ACC	Epochs	Training Time (s)	Test Time (s)
121	Adam		√		0.8028	0.7109	20	2202	129
122	AdamW		√		0.7700	0.7125	4	472	131
123	Adadelta		√		0.7584	0.7145	7	825	132
124	Adafactor		√		0.7457	0.7319	10	1170	131
125	Adam		√	√	0.8109	0.6992	17	2066	133
126	AdamW		√	√	0.8084	0.7054	4	487	132
127	Adadelta		√	√	0.7991	0.7070	9	1199	182
128	Adafactor		√	√	0.7926	0.6997	8	1056	167
129	Adam	√			0.7521	0.7283	12	1490	164
130	AdamW	√			0.7507	0.7300	4	500	169
131	Adadelta	√			0.7498	0.7358	6	749	165
132	Adafactor	√			0.7371	0.7315	5	635	179
133	Adam	√		√	0.7435	0.7213	13	1724	189
134	AdamW	√		√	0.7246	0.7315	12	1599	174
135	Adadelta	√		√	0.7044	0.7425	7	934	180
136	Adafactor	√		√	0.7118	**0.7472**	8	1059	187
137	Adam	√	√		0.7869	0.7324	12	1565	192
138	AdamW	√	√		0.7718	0.7217	4	528	182
139	Adadelta	√	√		0.7571	0.7285	8	1075	198
140	Adafactor	√	√		0.7541	0.7276	7	1478	216
141	Adam	√	√	√	0.7626	0.7210	12	2631	229
142	AdamW	√	√	√	0.7178	0.7337	12	2519	227
143	Adadelta	√	√	√	0.7238	0.7267	4	739	210
144	Adafactor	√	√	√	0.7320	0.7249	4	858	222
							Total	29,562	4219

**Table 10 bioengineering-10-01098-t010:** Results of the InceptionResNetV2 family.

ID	OPTZ	TL	TR	IA	Loss	ACC	Epochs	Training Time (s)	Test Time (s)
145	Adam		√		0.8118	0.7066	12	3164	269
146	AdamW		√		0.8171	0.6901	7	1887	267
147	Adadelta		√		0.8048	0.6975	7	1890	272
148	Adafactor		√		0.7941	0.7117	5	1369	283
149	Adam		√	√	0.8011	0.7165	15	4679	264
150	AdamW		√	√	0.8145	0.6922	4	1149	307
151	Adadelta		√	√	0.8067	0.6940	6	1744	343
152	Adafactor		√	√	0.7871	0.7076	8	3213	456
153	Adam	√			0.8422	0.6967	12	4939	455
154	AdamW	√			0.8247	0.6920	5	1838	418
155	Adadelta	√			0.8167	0.7019	11	4628	472
156	Adafactor	√			0.8269	0.6915	4	2113	547
157	Adam	√		√	0.8184	0.6886	16	8287	489
158	AdamW	√		√	0.7907	0.6997	11	4793	504
159	Adadelta	√		√	0.7863	0.7073	6	3332	1245
160	Adafactor	√		√	0.7823	0.7054	7	3881	1,211
161	Adam	√	√		0.8214	0.6962	16	4131	255
162	AdamW	√	√		0.8192	0.7011	9	2375	271
163	Adadelta	√	√		0.8087	0.7122	4	1047	262
164	Adafactor	√	√		0.7982	0.7057	6	1576	260
165	Adam	√	√	√	0.8305	0.6823	17	4530	265
166	AdamW	√	√	√	0.8246	0.6848	4	1095	313
167	Adadelta	√	√	√	0.8146	0.6906	9	2500	320
168	Adafactor	√	√	√	0.8136	0.6923	4	1133	330
							Total	71,296	10,077

**Table 11 bioengineering-10-01098-t011:** Results of the MobileNetV2 family.

ID	OPTZ	TL	TR	IA	Loss	ACC	Epochs	Training Time (s)	Test Time (s)
169	Adam		√		0.9017	0.6775	16	1209	76
170	AdamW		√		0.9028	0.6704	7	552	79
171	Adadelta		√		0.8984	0.6734	4	327	79
172	Adafactor		√		0.9001	0.6773	7	555	77
173	Adam		√	√	0.9104	0.6603	10	840	74
174	AdamW		√	√	0.9119	0.6839	7	592	81
175	Adadelta		√	√	0.9063	0.6779	9	833	84
176	Adafactor		√	√	0.8998	0.6809	5	467	83
177	Adam	√			0.8022	0.7108	11	935	85
178	AdamW	√			0.7965	0.7183	4	343	82
179	Adadelta	√			0.7940	0.7173	6	517	85
180	Adafactor	√			0.7759	0.7229	7	596	84
181	Adam	√		√	0.7684	0.7125	14	1304	85
182	AdamW	√		√	0.7910	0.7196	4	384	88
183	Adadelta	√		√	0.7795	0.7186	4	384	90
184	Adafactor	√		√	0.7715	0.7203	4	383	89
185	Adam	√	√		0.7996	0.7118	11	1016	104
186	AdamW	√	√		0.7855	0.7225	4	428	111
187	Adadelta	√	√		0.7622	0.7306	4	431	103
188	Adafactor	√	√		0.7712	0.7238	7	639	89
189	Adam	√	√	√	0.7532	0.7150	14	1350	90
190	AdamW	√	√	√	0.7434	0.7282	7	794	112
191	Adadelta	√	√	√	0.7261	**0.7345**	4	400	93
192	Adafactor	√	√	√	0.7353	0.7253	10	1166	113
							Total	16,445	2134

**Table 12 bioengineering-10-01098-t012:** Best results from each family.

ID	CNN	OPTZ	TL	TR	IA	Loss	ACC	Epochs	Training Time (s)	Test Time (s)
20	DenseNet201	Adafactor	√	√		0.7190	0.7610	6	2205	420
36	ResNet152V2	Adafactor	√			0.7413	0.7359	9	3771	452
68	NAYNetLarge	Adafactor	√	√		0.7813	0.7210	7	4260	768
85	VGG19	Adam	√		√	0.6276	**0.7628**	18	8930	163
111	Xception	Adadelta	√		√	0.6895	0.7499	6	1238	133
136	InceptionV3	Adafactor	√		√	0.7118	0.7472	8	1059	187
191	MobileNetV2	Adadelta	√	√	√	0.7261	0.7345	4	400	93

**Table 13 bioengineering-10-01098-t013:** Confusion matrix for the type of examination.

Real/Predicted	‘ct’	‘radiograph’	‘scan’	‘tomography’	‘xray’	‘angiogram’	’angiography’	‘abdominal’	‘contrast’
**‘ct’**	108	2	10	3	0	0	0	0	0
**‘radiograph’**	1	32	0	0	12	0	0	0	0
**‘scan’**	19	0	26	5	0	0	0	0	1
**‘tomography’**	16	0	9	0	0	0	0	0	0
**‘xray’**	0	0	0	0	11	0	0	0	0
**‘angiogram’**	1	0	0	0	0	9	1	0	0
**‘angiography’**	3	0	0	0	0	7	0	0	0
**‘abdominal’**	4	0	1	0	0	0	0	0	0
**‘contrast’**	0	0	0	0	0	3	0	0	0

**Table 14 bioengineering-10-01098-t014:** Metrics of the confusion matrix for the type of examination.

Class	P	N	TP	FP	TN	FN	TPR	TNR	PPV	NPV	ACC	F1
‘ct’	123	161	108	44	117	15	0.88	0.73	0.71	0.89	0.79	0.79
‘radiograph’	45	239	32	2	237	13	0.71	0.99	0.94	0.95	0.95	0.81
‘scan’	51	233	26	20	213	25	0.51	0.91	0.57	0.89	0.84	0.54
‘tomography’	25	259	0	8	251	25	0.00	0.97	0.00	0.91	0.88	-
‘xray’	11	273	11	12	261	0	1.00	0.96	0.48	1.00	0.96	0.65
‘angiogram’	11	273	9	10	263	2	0.82	0.96	0.47	0.99	0.96	0.60
‘angiography’	10	274	0	1	273	10	0.00	1.00	0.00	0.96	0.96	-
‘abdominal’	5	279	0	0	279	5	0.00	1.00	-	0.98	0.98	-
‘contrast’	3	281	0	1	280	3	0.00	1.00	0.00	0.99	0.99	-

**Table 15 bioengineering-10-01098-t015:** The confusion matrix of the body part.

Real/Predicted	‘chest’	‘abdomen’	‘artery’	‘lung’	‘lobe’	‘head’	‘liver’	‘pelvi’	‘bone’	‘pulmonary’	‘aortic’	‘vein’	‘hip’	‘neck’	‘femoral’
**‘chest’**	39	3	0	10	1	0	0	0	1	0	0	0	0	0	0
**‘abdomen’**	0	47	2	2	0	0	0	0	1	0	0	0	0	0	0
**‘artery’**	1	10	24	0	0	0	0	0	0	0	0	0	0	0	0
**‘lung’**	12	6	2	16	0	0	0	0	0	0	0	0	0	0	0
**‘lobe’**	0	22	1	1	1	0	0	0	0	0	0	0	0	0	0
**‘head’**	0	14	0	0	0	1	0	0	0	0	0	0	0	0	0
**‘liver’**	0	10	0	0	0	0	0	0	0	0	0	0	0	0	0
**‘pelvi’**	0	0	0	0	0	0	0	2	0	0	0	0	0	0	0
**‘bone’**	2	1	0	0	0	0	0	2	6	0	0	0	0	0	0
**‘pulmonary’**	2	0	1	4	0	0	0	0	0	0	0	0	0	0	0
**‘aortic’**	0	3	1	0	1	0	0	0	0	0	0	0	0	0	0
**‘vein’**	0	9	0	0	0	1	0	0	1	0	0	0	0	0	0
**‘hip’**	0	0	0	0	0	0	0	2	1	0	0	0	1	0	0
**‘neck’**	1	0	0	0	0	0	0	6	3	0	0	0	0	0	0
**‘femoral’**	0	0	0	0	0	0	0	1	5	0	0	0	0	0	1

**Table 16 bioengineering-10-01098-t016:** Metrics of the confusion matrix for the body parts.

Class	P	N	TP	FP	TN	FN	TPR	TNR	PPV	NPV	ACC	F1
**‘chest’**	54	230	39	18	212	15	0.72	0.92	0.68	0.93	0.88	0.70
**‘abdomen’**	52	232	47	78	154	5	0.90	0.66	0.38	0.97	0.71	0.53
**‘artery’**	35	249	24	7	242	11	0.69	0.97	0.77	0.96	0.94	0.73
**‘lung’**	36	248	16	17	187	20	0.44	0.92	0.48	0.90	0.85	0.46
**‘lobe’**	25	259	1	2	257	24	0.04	0.99	0.33	0.91	0.91	0.07
**‘head’**	15	269	1	1	268	14	0.07	1.00	0.50	0.95	0.95	0.12
**‘liver’**	10	274	0	0	274	10	0.00	1.00	-	0.96	0.96	-
**‘pelvi’**	2	282	2	11	271	0	1.00	0.96	0.15	1.00	0.96	0.27
**‘bone’**	11	273	6	12	261	5	0.55	0.96	0.33	0.98	0.94	0.41
**‘pulmonary’**	7	277	0	0	277	7	0.00	1.00	-	0.98	0.98	-
**‘aortic’**	5	279	0	0	279	5	0.00	1.00	-	0.98	0.98	-
**‘vein’**	11	273	0	0	273	11	0.00	1.00	-	0.96	0.96	-
**‘hip’**	4	280	1	0	280	3	0.25	1.00	1.00	0.99	0.99	0.40
**‘neck’**	10	274	0	0	274	10	0.00	1.00	-	0.96	0.96	-
**‘femoral’**	7	277	1	0	277	6	0.14	1.00	1.00	0.98	0.98	0.25

**Table 17 bioengineering-10-01098-t017:** Confusion matrix of the identified problem.

Real/Predicted	‘nodule’	‘hepatic’	‘aneurysm’	‘fracture’	‘normal’	‘tumor’	‘pancreatic’	‘effusion’	‘cystic’	‘hematoma’	‘cyst’
**‘nodule’**	30	3	2	0	0	4	0	4	8	0	0
**‘hepatic’**	0	5	1	1	0	0	16	0	22	0	0
**‘aneurysm’**	1	1	15	0	1	0	3	0	8	1	0
**‘fracture’**	0	0	0	30	3	0	0	0	1	0	0
**‘normal’**	0	0	9	1	21	0	0	0	1	0	0
**‘tumor’**	1	1	0	1	0	9	2	0	3	1	0
**‘pancreatic’**	0	2	0	0	0	0	15	0	2	3	0
**‘effusion’**	10	0	1	0	0	5	0	0	4	0	0
**‘cystic’**	0	0	2	0	0	0	6	2	9	0	0
**‘hematoma’**	0	1	0	0	0	0	3	0	3	0	0
**‘cyst’**	0	0	0	0	0	0	2	0	4	0	0

**Table 18 bioengineering-10-01098-t018:** Metrics of the confusion matrix for the identified problem.

Class	P	N	TP	FP	TN	FN	TPR	TNR	PPV	NPV	ACC	F1
**‘nodule’**	51	233	30	12	221	21	0.59	0.95	0.71	0.91	0.88	0.65
**‘hepatic’**	45	239	5	8	231	40	0.11	0.97	0.38	0.85	0.83	0.17
**‘aneurysm’**	30	254	15	15	239	15	0.50	0.94	0.50	0.94	0.89	0.50
**‘fracture’**	34	250	30	3	247	4	0.88	0.99	0.91	0.98	0.98	0.90
**‘normal’**	32	252	21	4	248	11	0.66	0.98	0.84	0.96	0.95	0.74
**‘tumor’**	18	266	9	9	257	9	0.50	0.97	0.50	0.97	0.94	0.50
**‘pancreatic’**	22	262	15	32	230	7	0.68	0.88	0.32	0.97	0.86	0.43
**‘effusion’**	20	264	0	6	258	20	0.00	0.98	0.00	0.93	0.91	-
**‘cystic’**	19	265	9	56	209	10	0.47	0.79	0.14	0.95	0.77	0.21
**‘hematoma’**	7	277	0	5	272	7	0.00	0.98	0.00	0.97	0.96	-
**‘cyst’**	6	278	0	0	278	6	0.00	1.00	-	0.98	0.98	-

## Data Availability

Publicly available datasets were analyzed in this study. This data can be found here: https://github.com/razorx89/roco-dataset (access on 18 September 2023).

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
