# Peer review of "A New Generative Model for Textual Descriptions of Medical Images Using Transformers Enhanced with Convolutional Neural Networks"

_bioengineering, 2023, doi:10.3390/bioengineering10091098_

Round 1

Reviewer 1 Report

The authors empirically evaluated a large number of models on the task of generating captions from medical images. The authors used the ROCO dataset, which contains multiple types of medical images including CT scans, and X-rays, among others. The authors conducted 192 experiments to evaluate the performance of the tested neural network models under different design conditions, including the optimization technique, the involvement of pre-training, image augmentation, the fine-tuning of the pre-trained weights, and the number of epochs. The authors reported multiple metrics including Accuracy, F1, Sensitivity, and specificity on the testing set in addition to the training and testing time.

The reviewer, however, raises the following points that may enhance the presentation and solidify the results,

  • The manuscript is too long. It does not read as a concise paper, but rather as a long book chapter. The authors can summarize the main points regarding the method, the results, and contributions. Some of the results can be moved to a supplementary file. It is hard to have 18 tables in the main manuscript.

  • The authors claimed that this work provides significant methodological and theoretical advancement to the field. The reviewer has to disagree with this claim. The current work only provides an empirical evaluation of multiple models under different conditions. 

  • While the authors discussed multiple models for feature extraction, very little was discussed about the models underlying the text generation component.

  • The authors focused on accuracy as a metric for selecting the best model while it is known that accuracy is not the best metric when it comes to imbalanced datasets.

  • Table 18 does not make a lot of sense since it compares different models for solving different problems.
  • The code used to generate the reported results is not available. The authors should share a reproducible version of the implementation.

Reviewer 2 Report

1. Describe the data set in a detailed table.

2. The resolution of Figure 1 and Figure 4 is too low

3. Describe the mathematical meaning of formula 1

4. Why choose the loss function of formula 2, please think about the explanation

5. Please describe the environment and hardware used in the experiment 

6. Add references from the last two years

Minor editing of English language required

Reviewer 3 Report

The manuscript proposes a new generative model that outputs text given a medical image as input. 

1. Figure 4 was apparently borrowed from the O. Pelka et al.'s paper. Please cite it or use a different image example from the ROCO dataset. 

2.  Consider adding a table that shows the numbers of (image, label) sets used for training, validation, and test for this study. How were the (image, label) sets obtained? Manually? or automatically?

3. Please shorten the results from the use of the pre-trained models (i.e., 3.1 - 3.8). It seems very redundant and occupies a lot of space in the paper. 

4. Figure 5 is not clear. The input is an image, and the output label is three words. But, the shown deep neural net model doesn't seem to output three words.  Isn't the proposed generative model based on a transformer model rather than multi-layer perceptron?

5. It would be interesting if the authors compare the results between a single pre-trained model's word classification and a transformer-based model's three-word generation by using confusion matrix.

6. A detailed description of the transformer model used for this study is necessary.

7. Please provide a link for the source code used for the study.   

Round 2

Reviewer 3 Report

1. It is hard to associate Figure 4 with Table 2.  Figure 4 does not seem to be very related to the problem that the authors address with the transformer model. It is helpful to add another figure that shows examples of images and their three-word labels listed in Table 2. 

2. Figure 6 needs to be improved by showing the details of the "Proposed Generative Model" part. It seems during testing the model takes an image as input and outputs three words, but the model box does not show details such as the use of a pre-trained model and a transformer model with three word outputs.  

3. Section 3.1 - 3.8 is too lengthy. These sections need to be moved to a supplemental material. 

Author Response

Letter to the reviewer is attached.
